# Epigenome-wide association analysis of infant bronchiolitis severity: a multicenter prospective cohort study

Zhaozhong Zhu [1] ✉, Yijun Li [2], Robert J. Freishtat [3,4,5], Juan C. Celedón [6], Janice A. Espinola [1], Brennan Harmon [3], Andrea Hahn [3,5,7], Carlos A. Camargo Jr [1], Liming Liang [2,8,9] & Kohei Hasegawa [1,9]

Bronchiolitis is the most common lower respiratory infection in infants, yet its pathobiology remains unclear. Here we present blood DNA methylation data from 625 infants hospitalized with bronchiolitis in a 17-center prospective study, and associate them with disease severity. We investigate differentially methylated CpGs (DMCs) for disease severity. We characterize the DMCs based on their association with cell and tissues types, biological pathways, and gene expression. Lastly, we also examine the relationships of severity-related DMCs with respiratory and immune traits in independent cohorts. We identify 33 DMCs associated with severity. These DMCs are differentially methylated in blood immune cells. These DMCs are also significantly enriched in multiple tissues (e.g., lung) and cells (e.g., small airway epithelial cells), and biological pathways (e.g., interleukin-1-mediated signaling). Additionally, these DMCs are associated with respiratory and immune traits (e.g., asthma, lung function, IgE levels). Our study suggests the role of DNA methylation in bronchiolitis severity.

Bronchiolitis—the most common lower respiratory infection among infants—is an important health problem[1]. While 30%–40% of infants develop clinical bronchiolitis, its severity ranges from a minor nuisance to a fatal infection[2,3]. Bronchiolitis is also the leading cause of hospitalization in U.S. infants, accounting for ~110,000 hospitalizations annually[4]. Approximately 5% of these infants undergo mechanical ventilation[4]. However, traditional risk factors (e.g., prematurity) do not sufficiently explain the differences in bronchiolitis severity[3], and its pathobiology remains to be elucidated. Our limited understanding of the disease mechanisms has hindered efforts to develop targeted treatment strategies in this large patient population.

Although bronchiolitis is caused by a viral infection, emerging evidence about its pathobiology suggests a complex interrelationship of environmental (e.g., viruses), genetic, and host immune factors[5-7]. Indeed, studies have reported associations of the transcriptome[8-10], proteome[9,11], metabolome[12-15], and microbiome[10,15-18] profiles with disease severity. However, these findings were unable to uncover the integrated contribution of infant's genetic makeup and environmental factors to the pathobiology of bronchiolitis. DNA methylation—a major type of epigenetic regulation—addresses this knowledge gap via characterizing cytosine-phosphate-guanine (CpG) sites that are a function of genetic-environmental interplay[19].

To address the knowledge gap in the literature, we aimed to investigate the role of the epigenome in bronchiolitis severity by applying epigenome-wide association study (EWAS) approaches to blood DNA methylation data from a multicenter prospective cohort of infants hospitalized for bronchiolitis.

[1]Department of Emergency Medicine, Massachusetts General Hospital, Harvard Medical School, Boston, MA, USA. [2]Department of Epidemiology, Harvard T.H.Chan School of Public Health, Boston, MA, USA. [3]Center for Genetic Medicine Research, Children's National Hospital, Washington, DC, USA. [4]Division of Emergency Medicine, Children's National Hospital, Washington, DC, USA. [5]Department of Pediatrics, George Washington University School of Medicine and Health Sciences, Washington, DC, USA. [6]Division of Pulmonary Medicine, Department of Pediatrics, UPMC Children's Hospital of Pittsburgh, University of Pittsburgh, Pittsburgh, PA, USA. [7]Division of Infectious Diseases, Children's National Hospital, Washington, DC, USA. [8]Department of Biostatistics, Harvard T.H.Chan School of Public Health, Boston, MA, USA. [9]These authors contributed equally: Liming Liang, Kohei Hasegawa. ✉e-mail: zzhu5@mgh.harvard.edu

## Results

Of the 1016 infants hospitalized for bronchiolitis enrolled into the 35th Multicenter Airway Research Collaboration (MARC-35) cohort, the current study examined 625 infants with high-quality blood DNA methylation data (Fig. 1 and Supplementary Fig. 1). The analytic and non-analytic cohorts did not differ in most patient characteristics ($P \geq 0.05$; Supplementary Table 2), except for several variables (e.g., age, race/ethnicity, respiratory syncytial virus (RSV) infection). Among the analytic cohort, the median age was 3 (interquartile range [IQR], 2–6) months, 38% were female, 46% were non-Hispanic White, 29% were Hispanic, and 22% were non-Hispanic Black. During hospitalizations for bronchiolitis, 5% of participants underwent positive pressure ventilation (PPV) (Table 1 and Supplementary Table 3). For DNA methylation profiling, a total of 863,904 CpGs were measured.

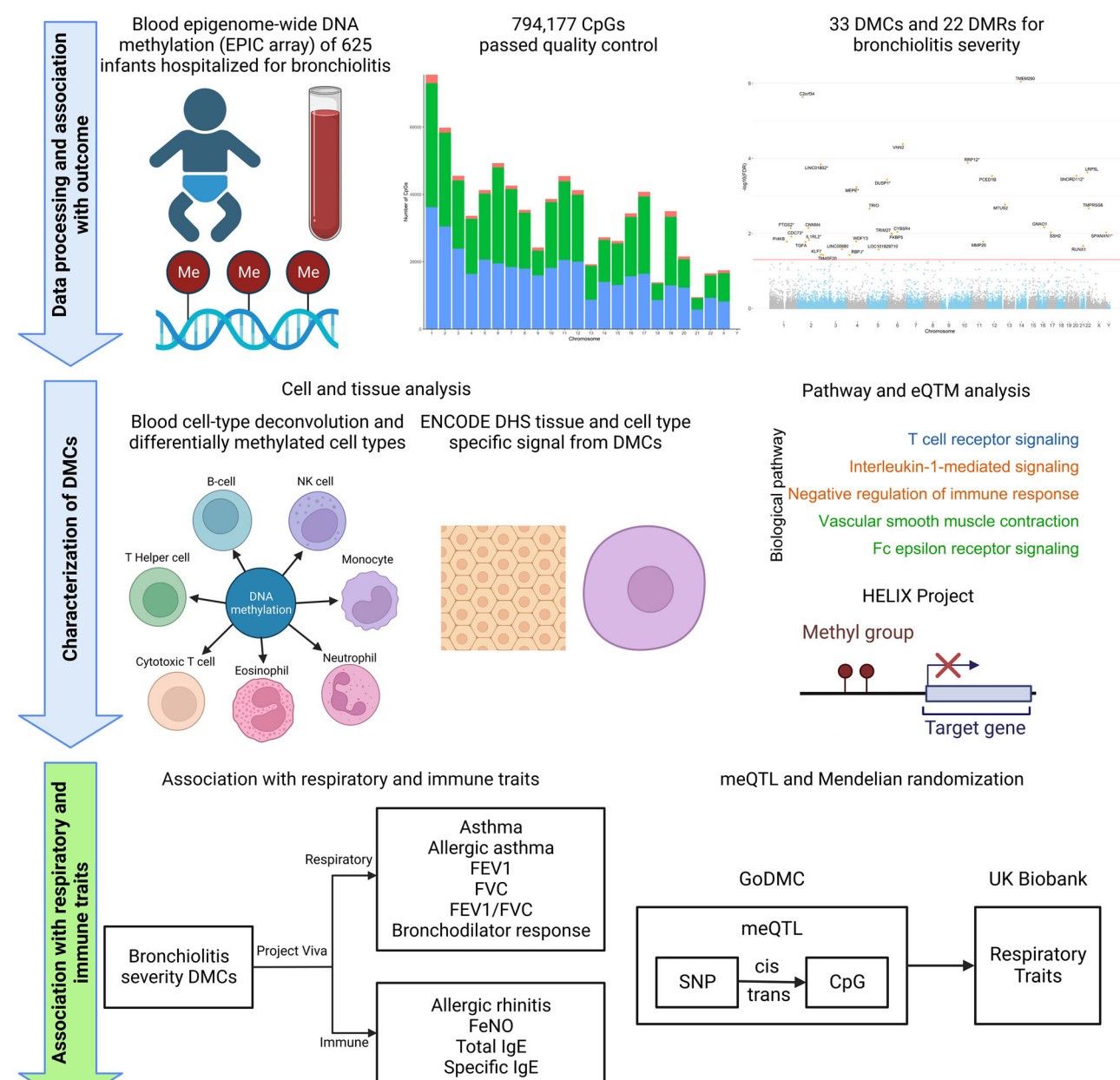

**Fig. 1 | Study Design and Analytic Workflow.** The analytical cohort consists of 625 infants hospitalized for bronchiolitis in a multicenter prospective cohort study—the 35th Multicenter Airway Research Collaboration (MARC-35). Blood Infinium MethylationEPIC array (850 K) DNA methylation data underwent quality control, leading to a total of 794,177 high-quality CpGs for the downstream analysis. In Aim 1, the association of 794,177 CpGs with the risk of PPV use was examined. A total of 33 severity-related DMCs and 22 DMRs were identified. In Aim 2, seven blood immune cell types were deconvoluted using the epigenome-wide DNA methylation data. The association of the DMCs with each cell type was examined. The ENCODE DHS tissue- and cell-specific signal from the DMCs was also determined. The biological pathway analysis using the GO, KEGG, and Reactome databases was performed. The association of blood DNA methylation and gene expression was investigated by cis-eQTM (HELIX Project) analysis. In Aim 3, by leveraging independent and publicly available EWAS (Project Viva) and GWAS (GoDMC and UK Biobank) data, the association of bronchiolitis severity-related DMCs with respiratory and immune traits was examined. Some components of this figure were created with BioRender.com. CpG, cytosine-phosphate-guanine; DHS, DNase hypersensitivity site; DMC, differentially methylated CpG; DMR, differentially methylated region; ENCODE, Encyclopedia of DNA Elements; eQTM, expression quantitative trait methylation; EWAS, epigenome-wide association study; GO, Gene Ontology; GoDMC, Genetics of DNA Methylation Consortium; GWAS, genome-wide association study; KEGG, Kyoto Encyclopedia of Genes and Genomes; PPV, positive pressure ventilation.

## Table 1 | Baseline characteristics and clinical course of 625 infants hospitalized for bronchiolitis

| Characteristics | Overall (*n* = 625) |
|---|---|
| **Demographics** | |
| Age (month), median (IQR) | 3 (2–6) |
| Female sex | 240 (38) |
| Race/ethnicity | |
| Non-Hispanic white | 287 (46) |
| Non-Hispanic black | 136 (22) |
| Hispanic | 180 (29) |
| Other or unknown | 22 (4) |
| Prematurity (32–36.9 weeks) | 107 (17) |
| Birth weight (kg), median (IQR) | 3.28 (2.90–3.60) |
| Mode of birth (cesarean delivery) | 210 (34) |
| Previous breathing problems before the index hospitalization[a] | |
| 0 | 488 (78) |
| 1 | 106 (17) |
| 2 | 31 (5) |
| Previous ICU admission | 8 (1) |
| History of eczema | 102 (16) |
| Lifetime antibiotic use | 201 (32) |
| Ever attended daycare | 153 (25) |
| Cigarette smoke exposure at home | 101 (16) |
| Maternal smoking during pregnancy | 101 (17) |
| Parental history of asthma | 204 (33) |
| Parental history of eczema | 122 (20) |
| **Clinical presentation** | |
| Weight at presentation (kg), median (IQR) | 6.20 (4.90–7.92) |
| Respiratory rate at presentation (per minute), median (IQR) | 48 (40–60) |
| Oxygen saturation at presentation | |
| <90% | 56 (9) |
| 90–93% | 99 (16) |
| ≥94% | 455 (75) |
| Blood eosinophilia (≥4%) | 60 (11) |
| IgE sensitization (%) | 128 (21) |
| Length of hospitalization (day), median (IQR) | 2 (1–3) |
| Corticosteroid use during hospitalization[b] | 82 (13) |
| **Respiratory virus** | |
| RSV infection | 473 (76) |
| RV infection | 110 (19) |
| **Acute clinical outcome** | |
| Positive pressure ventilation use[b] | 30 (5) |

Data are no. (%) of infants unless otherwise indicated. Percentages may not equal 100, because of rounding and missingness.

*ICU* intensive care unit, *IgE* immunoglobulin E, *IQR* interquartile range, *RSV* respiratory syncytial virus, *RV* rhinovirus.

[a]Defined as an infant having a cough that wakes him or her at night or causes emesis, or when the child has wheezing or shortness of breath without cough.

[b]Defined as the use of continuous positive airway pressure ventilation and/or mechanical ventilation during the hospitalization.

Among these, 794,177 CpGs passed stringent quality control and were included in the subsequent analysis (Supplementary Figs. 2 and 3).

### Epigenome-wide analysis demonstrated associations of CpGs and methylated regions with bronchiolitis severity

The EWAS results showed that the confounding and batch effects were well-controlled with minimal inflation ($\lambda_{\text{genomic control}} = 1.02$, Fig. 2A). A total of 33 differentially methylated CpGs (DMCs) were significantly associated with the risk of PPV use (false discovery rate [FDR] < 0.05),

with 27 (82%) being hypomethylated and six (18%) being hypermethylated (Table 2 and Fig. 2B). Of these DMCs, most were annotated to gene body (e.g., cg01680062 on *RUNX1*), transcription-start site (e.g., cg24346915 on *TMPRSS6*), or untranslated region (e.g., cg02936755 on *LRP5L*). In the stratified analysis within infants with RSV infection, 15 DMCs were significantly associated with PPV use (FDR < 0.05), with 13 being hypomethylated and two being hypermethylated (Supplementary Fig. 4A). Among infants with rhinovirus (RV) infection, three DMCs were significantly associated with PPV use (FDR < 0.05) and all being hypomethylated (Supplementary Fig. 4B). Additionally, in the region-based analysis, a total of 22 differentially methylated regions (DMRs) were significantly associated with the risk of PPV use (Šidák *p*-value < 0.05; Supplementary Table 4).

### Severity-related DMCs were associated with cell types, tissue types, biological pathways, and gene expression

Seven blood cells types were deconvoluted and inferred. The proportions of four cell types (helper T cells [(T$_H$)], monocytes, natural killer (NK) cells, and neutrophils) were significantly associated with the risk of PPV use (FDR < 0.05; Supplementary Table 5). Among them, neutrophils were the most strongly associated with the risk of PPV use (effect estimate = 0.13, FDR = $7.80 \times 10^{-5}$). The severity-related DMCs were also differentially methylated across blood immune cell types. There are a total of 51 significant DMC-cell pairs (FDR < 0.05), with 35 (69%) being hypomethylated, and 16 (31%) being hypermethylated (Fig. 3A). For example, cg02936755 on *LRP5L* was hypomethylated in cytotoxic T (T$_C$) cells (effect estimate = −0.72, FDR < 0.05) and T$_H$ cells (effect estimate = −0.36, FDR < 0.001); cg24346915 on *TMPRSS6* was hypermethylated in eosinophils (effect estimate = 1.00, FDR < 0.001; Fig. 3A). Among seven immune cell types, neutrophils had greatest number of associations with severity-related DMCs (23 out of 33). Integrative epigenomic analyses for PPV use highlighted the enrichment of DMCs with DNase hypersensitivity site (DHS) in various tissues (e.g., blood, lung) and related cell types (e.g., small airway epithelial cells, fetal lung fibroblasts; Fig. 3B). Finally, the gene-set enrichment analysis identified 5 pathways that were differentially enriched and related to respiratory and immune systems (FDR < 0.05; Fig. 3C), such as the T cell receptor signaling, interleukin-1 (IL-1)-mediated signaling, negative regulation of immune response and Fc epsilon receptor signaling pathways. Among the severity-related DMCs, we have identified 173 CpG-gene pairs from the blood-based cis-expression quantitative trait methylation (eQTM) data from the Human Early Life Exposome (HELIX) Project[20], of which one pair showed a significant association (cg12896170 and *TRIM27*, log2FC = −0.07, FDR = $2.39 \times 10^{-4}$; Supplementary Data 1).

### Severity-related DMCs were associated with respiratory and immune traits

Of 33 DMCs, fifteen were nominally-significantly associated with six respiratory and four immune traits in the independent and publicly available Project Viva study. For example, cg02534167 on *KLF7* was associated with allergic asthma, total immunoglobulin E (IgE) levels, specific IgE levels, and fractional exhaled nitric oxide (FeNO) level with a consistent direction of the effect. Additionally, cg07475825 on *CYB5R4* was associated with bronchodilator response (Fig. 4).

Based on the instrumental variable selection criteria, 26 independent SNPs for cg09541576, 43 SNPs for cg12547959, 244 SNPs for cg12896170, and 21 SNPs for cg15848159 were identified and used for Mendelian randomization analysis. The Mendelian randomization analysis suggested a significant relationship of four DMCs with asthma or lung function traits (FDR < 0.05). For example, cg12547959 was significantly associated with a higher risk of asthma (OR$_{\text{IVW}}$ = 1.02, 95% CI$_{\text{IVW}}$, 1.01−1.03, FDR$_{\text{IVW}}$ = $3.75 \times 10^{-6}$), reduced FEV1 level (effect estimate$_{\text{IVW}}$ = −0.014, 95% CI$_{\text{IVW}}$, −0.017, −0.012, FDR$_{\text{IVW}}$ = $1.43 \times 10^{-31}$), and reduced FVC level (effect

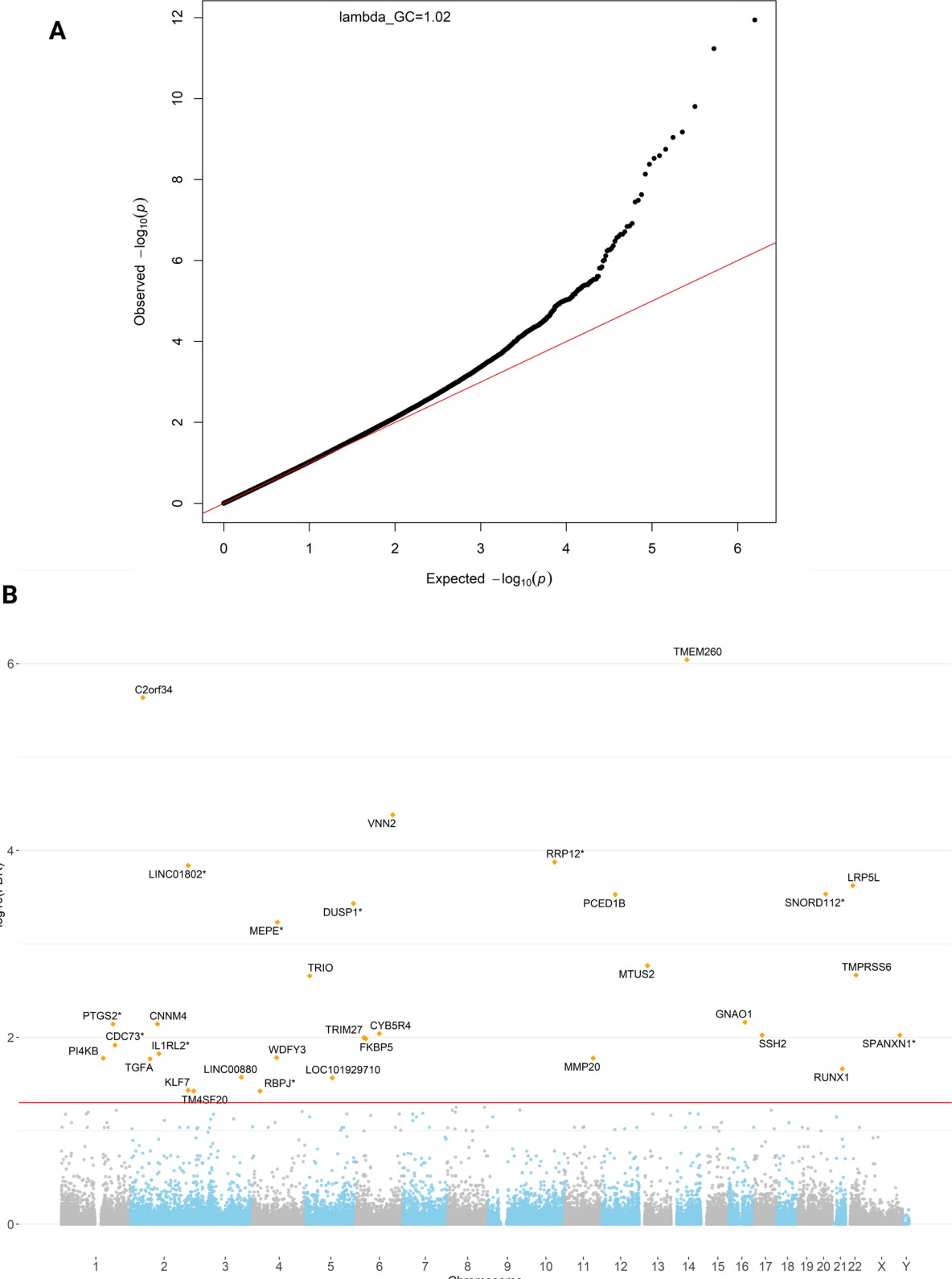

**Fig. 2 | Epigenome-wide Association of CpGs with Bronchiolitis Severity.**
**A** Association test quantile–quantile plot shows a departure from the null hypothesis of no association. Confounding and batch effects were well-controlled with minimal inflation ($\lambda_{\text{genomic control}} = 1.02$). The $\lambda_{\text{genomic control}}$ was calculated using *QCEWAS* package with default setting, which is based on one-sided Chi-Square test. **B** Manhattan plot for the epigenome-wide association test of bronchiolitis severity. The EWAS showed that a total of 33 DMCs were identified across 22 autosomal chromosomes and two sex chromosomes. The epigenome-wide significance level after accounting for multiple testing (FDR < 0.05) is denoted by the red line. DMC, differentially methylated CpG; EWAS, epigenome-wide association study; FDR, false discovery rate.

**Table 2 | Thirty-three severity-related CpG probes differentially methylated in infant hospitalized with bronchiolitis**

| CpG probe | Chromosome | Position | Effect estimate | P-value | FDR | Gene/nearest gene | Gene region feature category |
|---|---|---|---|---|---|---|---|
| cg03361294 | 14 | 57051003 | −0.58 | $1.14 \times 10^{-12}$ | $9.08 \times 10^{-7}$ | *TMEM260* | Body |
| cg09541576 | 2 | 44873248 | −0.31 | $5.83 \times 10^{-12}$ | $2.31 \times 10^{-6}$ | *C2orf34* | Body |
| cg15135194 | 6 | 133070524 | 0.44 | $1.57 \times 10^{-10}$ | $4.15 \times 10^{-5}$ | *VNN2* | Body |
| cg04175911 | 10 | 99172893 | −0.17 | $6.70 \times 10^{-10}$ | $1.33 \times 10^{-4}$ | *RRP12*[a] | 5'UTR[b] |
| cg05639088 | 2 | 208176244 | −0.33 | $9.14 \times 10^{-10}$ | $1.45 \times 10^{-4}$ | *LINC01802*[a] | TSS1500[b];3'UTR[b] |
| cg02936755 | 22 | 25771833 | −0.28 | $1.79 \times 10^{-9}$ | $2.37 \times 10^{-4}$ | *LRP5L* | 5'UTR |
| cg17555274 | 20 | 39122378 | −0.33 | $2.58 \times 10^{-9}$ | $2.93 \times 10^{-4}$ | *SNORD112*[a] | Unknown |
| cg25238420 | 12 | 47552760 | −0.29 | $2.98 \times 10^{-9}$ | $2.96 \times 10^{-4}$ | *PCED1B* | 5'UTR |
| cg20292908 | 5 | 172203421 | −0.42 | $4.19 \times 10^{-9}$ | $3.70 \times 10^{-4}$ | *DUSP1*[a] | 5'UTR[b] |
| cg27167895 | 4 | 88818173 | 0.29 | $7.36 \times 10^{-9}$ | $5.85 \times 10^{-4}$ | *MEPE*[a] | Unknown |
| cg16222694 | 13 | 29657404 | −0.29 | $2.36 \times 10^{-8}$ | $1.70 \times 10^{-3}$ | *MTUS2* | Body |
| cg24346915 | 22 | 37506589 | −0.45 | $3.25 \times 10^{-8}$ | $2.15 \times 10^{-3}$ | *TMPRSS6* | TSS1500 |
| cg12547959 | 5 | 14326153 | 0.40 | $3.60 \times 10^{-8}$ | $2.20 \times 10^{-3}$ | *TRIO* | Body |
| cg09432792 | 16 | 56352311 | 0.45 | $1.21 \times 10^{-7}$ | $6.88 \times 10^{-3}$ | *GNAO1* | Body |
| cg13132442 | 2 | 97464277 | −0.29 | $1.40 \times 10^{-7}$ | $7.20 \times 10^{-3}$ | *CNNM4* | Body |
| cg15002347 | 1 | 186590321 | −0.26 | $1.45 \times 10^{-7}$ | $7.20 \times 10^{-3}$ | *PTGS2*[a] | Unknown |
| cg07475825 | 6 | 84577581 | −0.37 | $1.95 \times 10^{-7}$ | $9.13 \times 10^{-3}$ | *CYB5R4* | Body |
| cg05790772 | X | 144114285 | −0.75 | $2.25 \times 10^{-7}$ | $9.47 \times 10^{-3}$ | *SPANXN1*[a] | Unknown |
| cg27459630 | 17 | 28019440 | −0.33 | $2.27 \times 10^{-7}$ | $9.47 \times 10^{-3}$ | *SSH2* | Body |
| cg12896170 | 6 | 28890069 | −0.29 | $2.54 \times 10^{-7}$ | $1.01 \times 10^{-2}$ | *TRIM27* | Body |
| cg00052684 | 6 | 35694245 | −0.37 | $2.73 \times 10^{-7}$ | $1.03 \times 10^{-2}$ | *FKBP5* | 5'UTR |
| cg08370869 | 1 | 193648305 | −0.28 | $3.35 \times 10^{-7}$ | $1.21 \times 10^{-2}$ | *CDC73*[a] | 5'UTR[b] |
| cg08552853 | 2 | 102875372 | −0.23 | $4.33 \times 10^{-7}$ | $1.50 \times 10^{-2}$ | *IL1RL2*[a] | Unknown |
| cg15848159 | 4 | 85791643 | 0.29 | $4.98 \times 10^{-7}$ | $1.65 \times 10^{-2}$ | *WDFY3* | 5'UTR |
| cg03574890 | 1 | 151284129 | −0.20 | $5.46 \times 10^{-7}$ | $1.67 \times 10^{-2}$ | *PI4KB* | Body;5'UTR |
| cg22885409 | 11 | 102470832 | −0.21 | $5.40 \times 10^{-7}$ | $1.67 \times 10^{-2}$ | *MMP20* | 3'UTR[b] |
| cg15920942 | 2 | 70743881 | −0.15 | $5.78 \times 10^{-7}$ | $1.70 \times 10^{-2}$ | *TGFA* | Body |
| cg01680062 | 21 | 36185960 | −0.23 | $7.66 \times 10^{-7}$ | $2.17 \times 10^{-2}$ | *RUNX1* | Body |
| cg20361768 | 3 | 156819083 | −0.27 | $9.76 \times 10^{-7}$ | $2.67 \times 10^{-2}$ | *LINC00880* | Body |
| cg01363387 | 5 | 95938059 | −0.20 | $1.02 \times 10^{-6}$ | $2.71 \times 10^{-2}$ | *LOC101929710* | Body |
| cg02534167 | 2 | 207987951 | 0.13 | $1.44 \times 10^{-6}$ | $3.68 \times 10^{-2}$ | *KLF7* | Body |
| cg03489069 | 2 | 228236715 | −0.26 | $1.56 \times 10^{-6}$ | $3.75 \times 10^{-2}$ | *TM4SF20* | Body |
| cg09412707 | 4 | 26085653 | −0.24 | $1.55 \times 10^{-6}$ | $3.75 \times 10^{-2}$ | *RBPJ*[a] | Unknown |

*CpG* cytosine-phosphate-guanine, *FDR* false discovery rate, *TSS* transcription-start site, *UTR* untranslated region.
[a]No gene is mapped based on CpG location, nearest genes are shown.
[b]Annotated based on the GENCODE v12 database. All others are annotated based on University of California, Santa Cruz (UCSC) RNA reference sequences collection (RefSeq).

estimate$_{IVW}$ = −0.014, 95% CI$_{IVW}$, −0.017, −0.011, FDR$_{IVW}$ = $9.66 \times 10^{-23}$; Fig. 5). Most of the associations were consistent across three Mendelian randomization methods (Fig. 5).

## Discussion

By applying the EWAS approach to data from a multicenter prospective cohort of infants hospitalized with bronchiolitis, we identified 33 CpGs differentially methylated in relation to the risk of PPV use. Furthermore, we observed that these DMCs were differentially methylated in blood immune cells—e.g., T$_C$ cells, T$_H$ cells, and neutrophils. These DMCs were also significantly and differentially enriched across multiple tissues (e.g., blood, lung) and cells (e.g., small airway epithelial cells, fetal lung fibroblasts), and biological pathways—e.g., T cell receptor signaling, IL-1-mediated signaling, and Fc epsilon receptor signaling pathways. Moreover, by leveraging publicly available EWAS data, the severity-related DMCs were associated with respiratory and immune traits (e.g., asthma, total IgE levels). Finally, we identified that four DMCs that were associated with asthma risk and lung function. Our EWAS study that has demonstrated the potential role of DNA methylation in the pathobiology of infant bronchiolitis—a major health problem.

Concordant with the present study, a growing body of evidence supports the relationship of DNA methylation with respiratory outcomes, such as asthma[21–31], chronic obstructive pulmonary disease (COPD)[32–35], idiopathic pulmonary fibrosis[36], lung function[32,33,37,38], and respiratory viral infection[39–42]. For example, a *post hoc* analysis from the MAKI study—a randomized, placebo-controlled trial of RSV immunoprophylaxis in preterm infants in the Netherlands—has reported three differentially methylated CpGs in nasal cells at age 6 years[40]. Furthermore, in a cohort study of 77 infants with RSV infection in Spain, blood DNA methylation signatures at infancy were associated with a higher risk of chronic respiratory sequelae, such as recurrent wheeze and asthma[41]. In addition, patients who developed respiratory sequelae showed a significantly higher proportion of T$_C$ and NK cells[41]. The current study—with a sample size many times larger than any other prior study on acute respiratory infection among infants—corroborates these earlier findings and extends them by demonstrating novel blood DNA methylation signatures in infants hospitalized with bronchiolitis and their relationship with acute disease severity and additional respiratory and immune related traits.

There are several potential mechanisms linking DNA methylation with bronchiolitis severity and its respiratory sequelae. First,

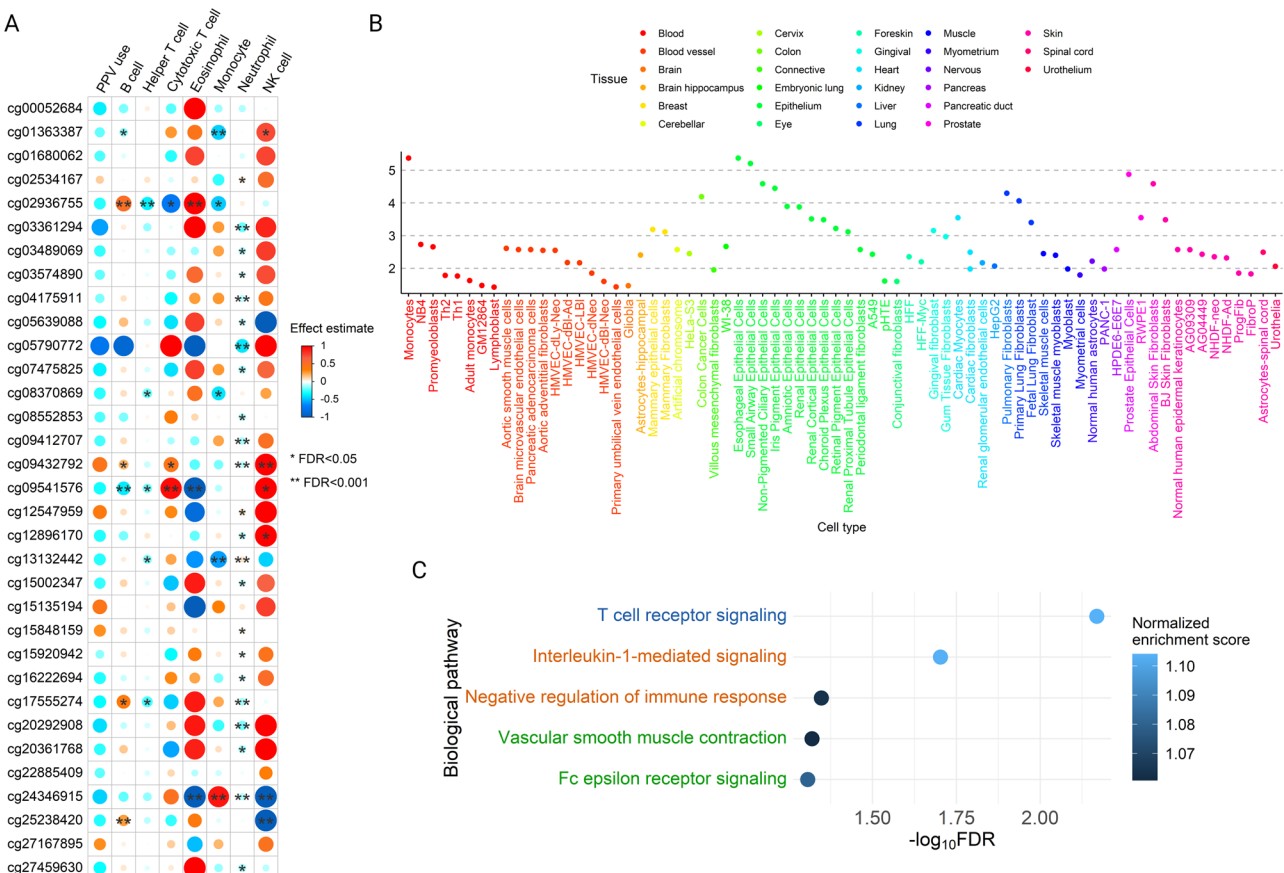

**Fig. 3 | Association of Severity-related Differentially Methylated CpGs with Different Tissue Types, Cell Types, and Biological Pathways. A** Blood cell type deconvolution analysis inferred seven blood immune cell types, including B cells, $T_H$ cells, $T_C$ cells, eosinophils, monocytes, neutrophils, and NK cells. After estimating cell type fractions, we identified that the DMCs were differentially methylated (hypermethylation or hypomethylation) in these cell types. The first column "PPV use" represents the overall effect size for PPV use (i.e., non-deconvoluted). The size of the dot denotes the magnitude of the associations. One asterisk denotes FDR < 0.05; two asterisks denote FDR < 0.001. **B** Enrichment of DMCs in DHS elements from the ENCODE Project. The DMCs showed significant enrichment in a

total of 73 cell types from 33 tissue types (FDR < 0.05). **C** Biological pathway analysis using GO, KEGG, and Reactome databases. We identified 5 respiratory or immune related differentially enriched pathways associated with bronchiolitis severity (FDR < 0.05). Blue color denotes Reactome pathways, orange color denotes KEGG pathways, and green color denotes GO biological process pathways. *DHS* DNase hypersensitivity site, *DMC* differentially methylated CpG, *ENCODE* Encyclopedia of DNA Elements, *FDR* false discovery rate, *GO* Gene ontology, *KEGG* Kyoto encyclopedia of genes and genomes, *NK* natural killer, *PPV* positive pressure ventilation, $T_C$ cytotoxic T, $T_H$ helper T.

the literature has suggested the role of host immune response—e.g., type I interferons (IFN), neutrophils—in the bronchiolitis pathobiology and viral respiratory infection. DNA methylation, as mediator between respiratory virus infections and disease severity, modulate airway and systematic inflammatory processes[42,43]. For example, a recent study has identified that blood DNA methylation signatures were associated with the activation of $T_C$ cells, neutrophils, and IFN signaling pathway in patients with severe SARS-CoV-2 infection[42]. Concordant with these findings, the current study found that the severity-related DMCs were differentially methylated in circulating immune cells, especially in $T_C$ cells, $T_H$ cells and neutrophils. Such differential methylation supported the heterogenous effect of the severity-related DMCs across blood immune cells. For example, cg02936755 on *LRP5L* was hypomethylated in $T_C$ cells, $T_H$ cells, and monocytes; however, it was hypermethylated in B cells and eosinophils. Furthermore, the DMCs were also significantly enriched in the T cell receptor signaling and type I IFN production pathways. Of note, previous research has reported that these pathways have been associated with bronchiolitis severity[10] and asthma development[5,44–46]. These studies have also shown that the regulation of these pathways is being mediated by epigenetic changes at the promoter level of the implicated genes[47].

The enrichment of DMCs with DHS regulatory elements in various tissues (e.g., blood, lung) and related cell types (e.g., small airway epithelial cells, fetal lung fibroblasts) supports that our findings in the blood can inform functional implications in the respiratory system. For example, a recent EWAS meta-analysis study of blood samples has identified that 1267 CpGs (1042 implicated genes) in blood were differentially methylated in relation to lung function[38]. Multiple implicated genes from the EWAS meta-analysis study are also identified from our study, such as *FKBP5* and *TGFA*, indicating the common role of blood DNA methylation in the respiratory system. Furthermore, some of our severity-related DMCs implicated genes (both within and nearby) play important roles in inflammation and immunity in the lung. For example, an in vitro study has found that dual-specificity phosphatase 1 (*DUSP1*) promotes virus-induced apoptosis and suppresses cell migration in RSV-infected epithelial cells. These processes further prevent dephosphorylation of c-Jun N-terminal kinase (JNK) and p38 mitogen-activated protein kinase (MAPK) as well as downstream cytokine production[48].

Lastly, the role of severity-related DMCs on respiratory sequelae warrants clarification. The current study has identified that cg01680062 on *RUNX1* and cg08552853 near *IL1RL2* are significantly associated with bronchiolitis severity. A previous study has shown that

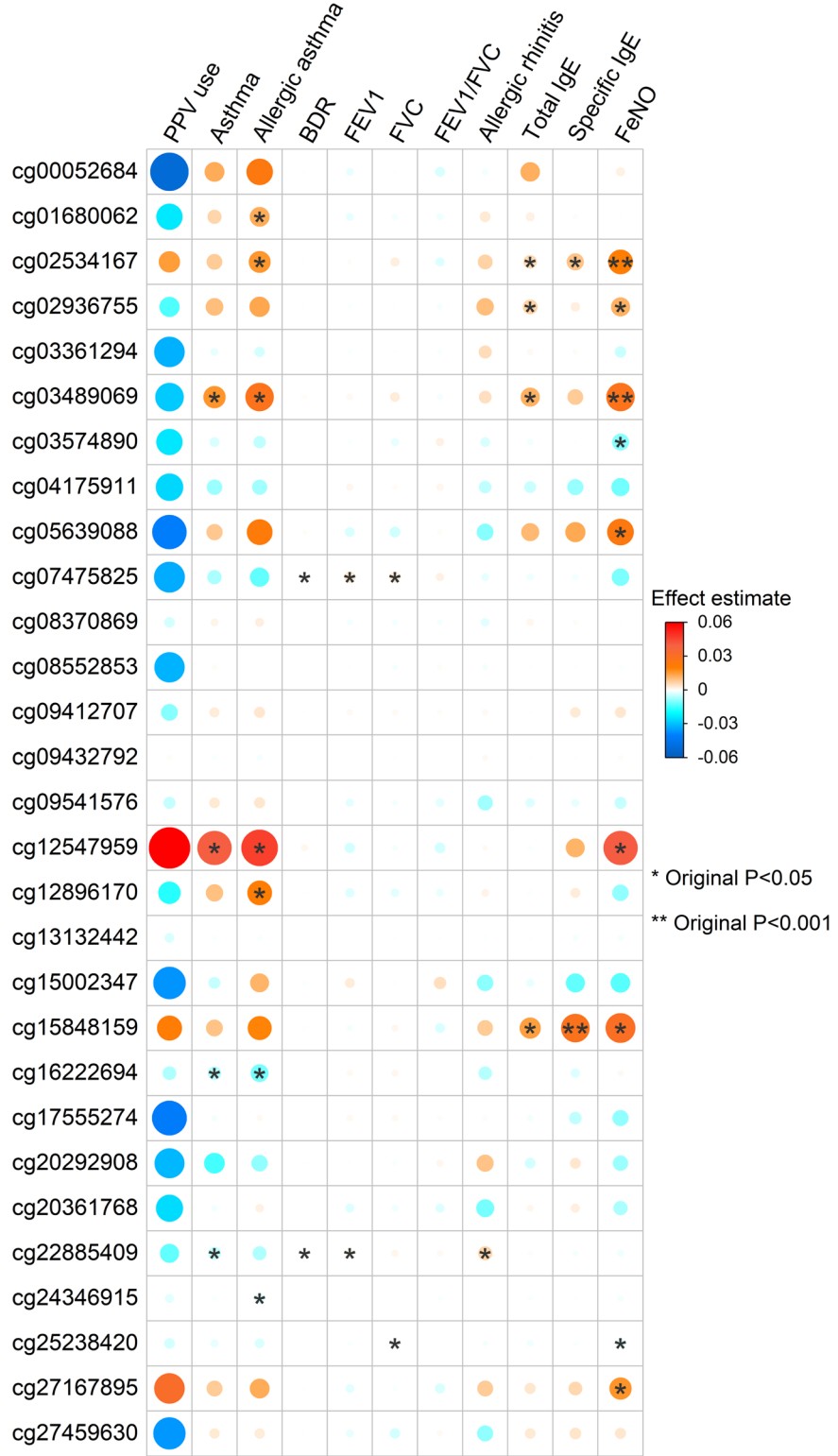

**Fig. 4 | Association of Severity-related Differentially Methylated CpGs with Respiratory and Immune Traits.** EWAS summary statistics for six respiratory (asthma, allergic asthma, FEV1, FVC, FEV1/FVC, and BDR) and four immune (allergic rhinitis, FeNO, total IgE, and specific IgE) traits from the independent and publicly available Project Viva study have been retrieved. The first column has shown the 33 DMCs' effect size from PPV use in MARC-35 study, we have calculated the effect size based on β-value to match the magnitude of effect sizes from Project Viva since they were also calculated based on β-value. This column will be helpful to compare the direction of effects for PPV use and the other traits. All other columns are from Project Viva study. Of 33 DMCs, 15 were nominally significant across ten traits ($P < 0.05$). The analysis was not adjusted for multiple comparison. The size of the dot denotes the magnitude of the associations. One asterisk denotes $P < 0.05$; two asterisks denote $P < 0.001$. *BDR* bronchodilator response, *FeNO* fractional exhaled nitric oxide, *FEV1* forced expiratory volume in one second, *FVC* forced vital capacity, *IgE* immunoglobulin E, *PPV* positive pressure ventilation.

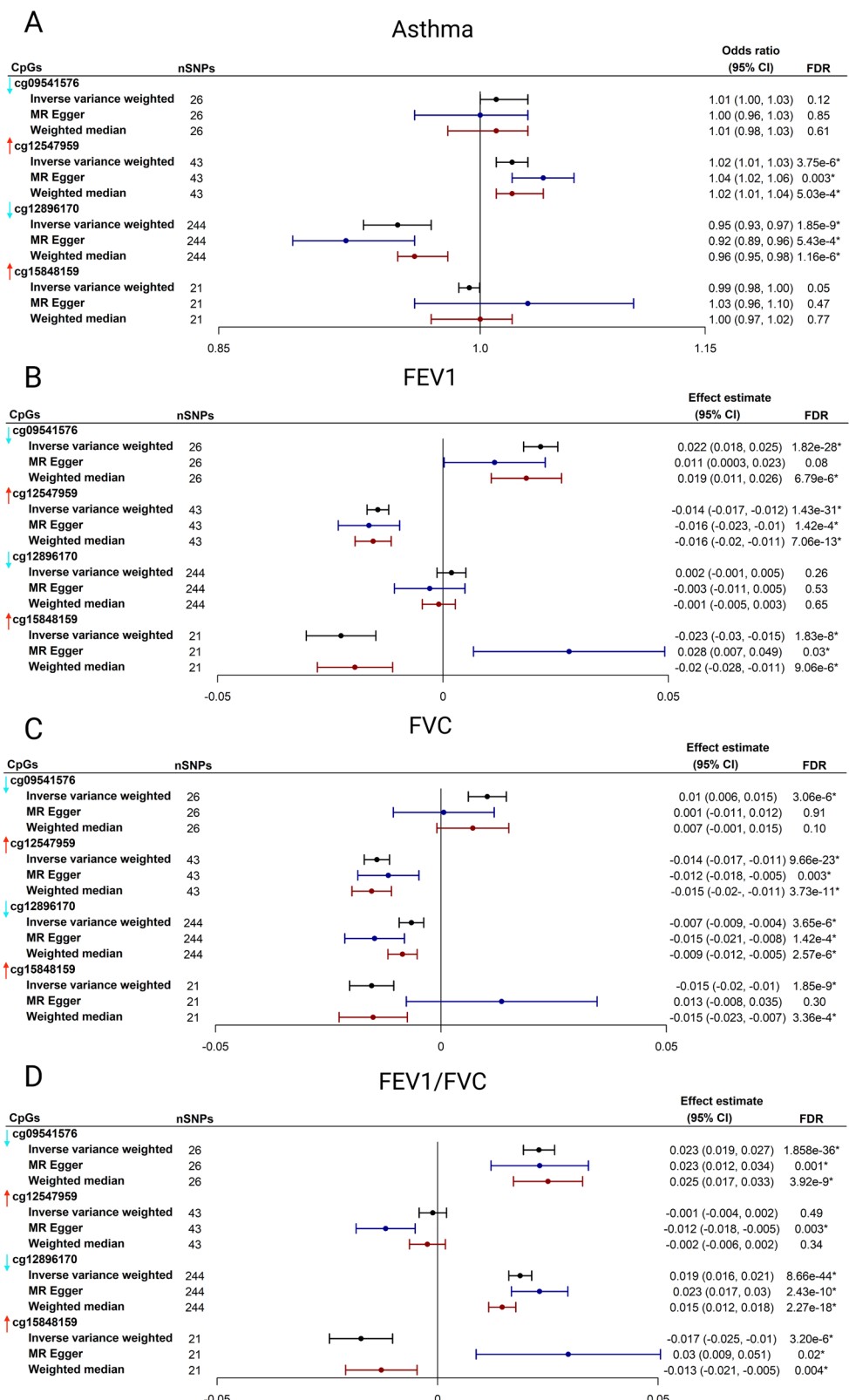

intrauterine smoke exposure decreased *RUNX1* expression during postnatal period[49]. Genetic variation in *RUNX1* is associated with airway responsiveness in children with asthma, and the association is modified by intrauterine smoke exposure[49]. Our previous large-scale genome-wide association study (GWAS) has identified *IL1RL2* as a pleotropic gene that is shared between allergic diseases and asthma[50]. *IL1RL2* encodes a cytokine receptor that belongs to the IL-1 receptor family[51]. Studies have shown a lower expression of *IL1RL2* in asthma causing increased IL-1 activity due to the lack of adequate anti-inflammatory regulation[52]. Finally, our Mendelian randomization analysis has found that four DMCs were associated with asthma risk and lung function, which potentially shows bronchiolitis severity and respiratory sequelae being common consequences of epigenetic regulatory impacts of the genetic variants. Of note, the results from

**Fig. 5 | Mendelian Randomization Analysis of Severity-related Differentially Methylated CpGs .** Mendelian randomization analysis was performed to investigate the relationships between four CpGs (cg09541576, cg12547959, cg12896170, and cg15848159) and four respiratory traits (**A**: asthma; **B**: FEV1; **C**: FVC, and **D**: FEV1/FVC). The vertical arrow on the left side of each CpG represents the direction of effect for PPV use. The arrows will be helpful to compare the direction of effects for PPV use and the four respiratory traits. Red arrow denotes hypermethylation (i.e., CpG was positively associated with PPV use). Aqua arrow denotes hypomethylation (i.e., CpG was negatively associated with PPV use). The meQTL data were retrieved from the GoDMC and respiratory traits. The GWAS data were retrieved from UK Biobank. Three Mendelian randomization approaches were used, including inverse variance-weighted method, MR–Egger regression method, and weighted median method. One asterisk denotes FDR < 0.05 after accounting for the multiple testing in the Mendelian randomization analysis. The center for the error bars denotes effect estimate. CpG, cytosine-phosphate-guanine; FEV1, forced expiratory volume in one second; FVC, forced vital capacity; GoDMC, Genetics of DNA Methylation Consortium; meQTL, methylation quantitative trait loci; GWAS, genome-wide association study; MR, Mendelian randomization; PPV, positive pressure ventilation.

Mendelian randomization were also consistent with the results from the independent Project Viva study—e.g., cg12547959 on *TRIO* was associated with higher asthma risk and reduced FEV1. Consistently, a recent large-scale GWAS of asthma and COPD overlap has identified a highly significant chromatin interaction in fetal lung fibroblasts overlapping with *TRIO*[53]. Notwithstanding the complexity of these mechanisms, the identification of the relationship between DNA methylation and bronchiolitis severity is important. Evidence has suggested that DNA methylation can be targeted for epigenetic therapy[54]. Our findings, in conjunction with the existing literature, should advance research into the development of DNA methylation-based strategies for bronchiolitis treatment and primary prevention of its respiratory sequelae.

Our study has several potential limitations. First, the cross-sectional design limited us to investigate the exact causal link between the DNA methylation signature and bronchiolitis severity. Second, although our Mendelian randomization analysis showed the association of severity-related DMCs in infancy with respiratory outcomes in later life (e.g., asthma and lung function), it is important to investigate the association of these CpGs in infancy with respiratory outcomes in later life in a longitudinal design[27,37]. Third, blood samples were used for DNA methylation profiling, which limited our inference to other tissue types (e.g., airway). Fourth, although we have used the cis-eQTM data from the HELIX Project to investigate the association of CpGs and gene expression, the current study lacks paired transcriptome data in blood to investigate the effect of DNA methylation on gene expression. Fifth, the results of DMCs in each cell type need to be interpreted with caution since "CellDMC" function in the *EpiDISH* package assumes all other cell types are 0% when it estimates a specific cell type driving the methylation change, where our data contain mixed cell types. Sixth, while nearly half of the identified CpGs were associated with respiratory and immune traits in an independent study, our inferences warrant external replication using the same bronchiolitis severity outcome. However, to our best knowledge, DNA methylation data with the same outcome are not currently available. Seventh, the current study did not have mechanistic experiments to validate the identified CpG functions. Yet, this study derives well-calibrated hypotheses that facilitate future experiments. Lastly, despite the study sample consisting of racially/ethnically- and geographically-diverse infants, our inferences must be cautiously generalized beyond infants hospitalized with bronchiolitis. Nonetheless, our data remain directly relevant for the 110,000 infants hospitalized yearly in the U.S[4].

In conclusion, by applying EWAS approach to a multicenter cohort of infants hospitalized with bronchiolitis, we identified that blood DNA methylation signatures were associated with bronchiolitis severity and played important roles in tissues, cells, pathways, and gene expression. For example, the severity-related CpGs were differentially methylated in blood immune cells, including $T_C$ cells, $T_H$ cells and neutrophils; and enriched in T cell receptor signaling pathway and IL-1-mediated signaling pathways. Additionally, these CpGs were associated with additional respiratory and immune traits, such as asthma, lung function, FeNO, and total IgE levels in an independent

and publicly available study. Our findings should facilitate further research into the interplay between environmental factors, epigenetics, host response, and disease pathobiology of infant bronchiolitis. This will, in turn, advance the development of targeted therapeutic measures (e.g., modification of DNA methylation-related immune response) and help clinicians manage this population with a large morbidity burden.

## Methods

### Study design, setting, and participants
The study design and analytic workflow are summarized in Fig. 1. We analyzed data from a multicenter prospective cohort study of infants hospitalized for bronchiolitis—the MARC-35 study[15,16]. Details of the study design, setting, participants, data collection, testing, and statistical analysis may be found in the *Supplementary Methods*. At 17 medical centers across 14 U.S. states (Supplementary Table 1), MARC-35 enrolled infants (age <1 year) who were hospitalized with an attending physician diagnosis of bronchiolitis during three bronchiolitis seasons in 2011–2014. The diagnosis of bronchiolitis was made according to the American Academy of Pediatrics bronchiolitis guidelines, defined as an acute respiratory illness with a combination of rhinitis, cough, tachypnea, wheezing, crackles, or retraction[55]. We excluded infants with preexisting heart or lung disease, immunodeficiency, immunosuppression, or gestational age of <32 weeks. All infants were managed at the discretion of the treating physicians. Of 1016 infants enrolled in the MARC-35 cohort, the current study investigated 625 infants with high-quality blood DNA methylation data (Supplementary Fig. 1). The institutional review board at each participating hospital approved the study with written informed consent obtained from the parent or guardian.

### Data collection and exposure
Clinical data (study participants' demographic characteristics, family, environmental, medical history, and details of the acute illness) were collected via structured interview and chart reviews using a standardized protocol[55,56]. After the index hospitalization for bronchiolitis, trained interviewers began interviewing parents/legal guardians by telephone at 6-month intervals in addition to medical record review by physicians. All data were reviewed at the Emergency Medicine Network Coordinating Center at Massachusetts General Hospital (Boston, MA, USA)[56]. Whole blood specimens were collected within 24 h of hospitalization using a standardized protocol[14]. The details of the data collection and measurement methods are described in the *Supplementary Methods*.

**Blood DNA methylation profiling and quality control.** The details of DNA extraction, DNA methylation profiling, and quality control are described in *Supplementary Methods*. Briefly, after DNA extraction, we performed DNA methylation profiling using the Illumina Infinium MethylationEPIC BeadChip (Illumina, San Diego, CA). To ensure the quality of the DNA methylation data, we followed the existing data preprocessing pipeline in the *minfi* package[57]. We applied multiple

sample-level and probe-level quality control filters (Supplementary Figs. 1 and 2 and *Supplementary Methods*). Following the quality control steps, we applied the single-sample normal-exponential normalization using the out-of-band probes (ssNoob) procedure to conduct background correction and dye bias correction[58].

## Outcome

The outcome of interest was higher disease severity defined by the use of PPV (i.e., continuous positive airway pressure and/or intubation with mechanical ventilation) during the hospitalization for bronchiolitis[12].

## Statistical analysis

The analytic workflow is summarized in Fig. 1. First, to investigate the relationship of the CpGs with the risk of PPV use, we performed EWAS analysis using linear regression models implemented by the *Meffil* package[59]. We used the empirical Bayes approach to obtain a robust estimation of standard error for the coefficients. To fit the linear regression model with normally distributed dependent variable (i.e., CpGs), we logit-transformed β-values to M-values. We used M-values for each CpG as the dependent variable in the association model. To account for the effects of technical batch and unknown confounding effect, we conducted a surrogate variable analysis by using *SmartSVA* package[60]. In the EWAS analysis, we adjusted for potential confounders, including age, sex, race/ethnicity, number of previous breathing problems, RSV infection, prematurity, seven blood cell types (B cells, $T_C$ cells, $T_H$ cells, eosinophils, monocytes, neutrophils, and NK cells), and the derived surrogate variables based on a priori knowledge and clinical plausibility[3,10]. Based on a priori-defined hypothesis[3], we also repeated the EWAS analysis stratified by RSV and RV infection. We corrected multiple testing using the Benjamini-Hochberg FDR method[61]. We defined DMCs as those CpGs significantly associated with PPV use at an FDR < 0.05. To identify the DMRs associated with PPV use, we applied the *comb-p* method[62] to the EWAS result. Specifically, the following parameters were used in the *comb-p* method to identify DMRs: (1) window size of 1 kb (--dist 1000); (2) minimum *p*-value of 0.01 to start a region (--seed 0.01); (3) Šidák *p*-value less than 0.05; and (4) at least 3 CpGs in the region. The annotations of the DMRs, including the nearest gene and transcript, were obtained from the UCSC genome browser (hg19).

Second, we performed blood cell type deconvolution analysis. We inferred seven blood cell types, including B cells, $T_C$ cells, $T_H$ cells, eosinophils, monocytes, neutrophils, and NK cells from our DNA methylation data using *EpiDISH* package[63]. We used β-value as the input for this analysis based on the package default settings. After estimating cell type fractions, we investigated the association of seven cell types with the risk of PPV use and whether the DMCs are differentially methylated (i.e., hypermethylation or hypomethylation) in these cell types. We also investigated the enrichment of the DMCs in DHS regulatory elements from the Encyclopedia of DNA Elements (ENCODE) Project[64] across 33 tissue types and 117 cell types using eFORGE 2.0[65]. We performed biological pathway analysis based on Gene Ontology (GO), Kyoto Encyclopedia of Genes and Genomes (KEGG), and Reactome pathways by using *methylGSA* package[66]. We investigated the association of the DMCs with transcription of nearby genes using publicly available blood-based cis-eQTM data from 823 European ancestry children in the HELIX Project[20]. The detail of this dataset is described in *Supplementary Methods*.

Third, by leveraging publicly available EWAS and GWAS data, we investigated the association of severity-related DMCs with respiratory and immune traits. We retrieved the EWAS summary statistics of six respiratory (asthma, allergic asthma, FEV1, FVC, FEV1/FVC, and bronchodilator response) and four immune (allergic rhinitis, FeNO, total IgE levels, specific IgE levels) traits from the an independent and publicly available Project Viva study by Cardenas and colleagues[25], and examined the association of the DMCs with these traits. The Project

Viva study collected nasal swabs from the anterior nares of 547 children (mean age 12.9 year) and measured DNA methylation with the Infinium MethylationEPIC BeadChip[25]. We also examined the methylation quantitative trait loci (meQTL) for the DMCs using a publicly available dataset from Genetics of DNA Methylation Consortium[67]. Finally, we performed Mendelian randomization analysis to investigate the potential causal relationships of severity-related DMCs (meQTL data from the Genetics of DNA Methylation Consortium) with four respiratory traits (GWAS data from the UK Biobank), including asthma[50,68-71], FEV1[72], FVC[72], and FEV1/FVC[72]. The details of these datasets and MR analysis are described in *Supplementary Methods*.

## Reporting summary

Further information on research design is available in the Nature Portfolio Reporting Summary linked to this article.

## Data availability

The EWAS summary statistics generated in this study are available at http://lianglab.rc.fas.harvard.edu/BronchiolitisSeverityEWAS/. In addition, the raw data that support the findings of this study will be available on the NIH/NIAID ImmPort under Accession ID: SDY2306 through controlled access to be compliant with the informed consent forms of MARC-35 study and the genomic data sharing plan. All other data are publicly available through the original studies' website. Project Viva data are available at https://figshare.com/articles/dataset/The_Nasal_Methylome_as_a_Biomarker_of_Asthma_and_Airway_Inflammation_in_Children/8285612/1. GoDMC data are available at http://mqtldb.godmc.org.uk/. UK Biobank data are available at https://www.ebi.ac.uk/gwas/. GENCODE data are available at https://www.gencodegenes.org/. UCSC RefSeq data are available at https://genome.ucsc.edu/cgi-bin/hgTrackUi?g=refGene.

## Code availability

The EWAS analysis was performed using linear regression models implemented using the *Meffil* package https://github.com/perishky/meffil. The region-based analysis was performed using *comb-p* method https://github.com/brentp/combined-pvalues. The blood cell type deconvolution analysis was performed using the *EpiDISH* package https://bioconductor.org/packages/release/bioc/html/EpiDISH.html. The DHS enrichment analysis was performed using eFORGE 2.0 https://eforge.altiusinstitute.org/. The pathway analysis was performed using *methylGSA* package https://bioconductor.org/packages/release/bioc/html/methylGSA.html. The GWAS analysis was performed using BOLT-LMM v2.3 https://alkesgroup.broadinstitute.org/BOLT-LMM/BOLT-LMM_manual.html. The Mendelian randomization analysis was performed using *TwoSampleMR* package https://mrcieu.github.io/TwoSampleMR/.

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

## Acknowledgements

This study was supported by grants from the National Institutes of Health (Bethesda, MD): K01 AI-153558 (Z.Z.), U01 AI-087881 (C.A.C.), R01 AI-114552 (C.A.C.), R01 AI-127507 (C.A.C. and R.J.F.), R01 AI-134940 (K.H.), R01 AI-137091 (K.H.), R01 AI-148338 (L.L. and K.H.), and UG3/UH3 OD-023253 (C.A.C.); Massachusetts General Hospital Department of Emergency Medicine Fellowship/Eleanor and Miles Shore Faculty Development Awards Program (Z.Z.); and the Harvard University William F. Milton Fund (Z.Z.). The content of this manuscript is solely the responsibility of the authors and does not necessarily represent the official views of the National Institutes of Health. We would like to thank the participants and researchers from the Multicenter Airway Research Collaboration (MARC) who significantly contributed or collected data. We thank Wade O'Brien and Aszia Burrell for their assistance with specimen processing. We also thank Dr. Michihito Kyo for his assistance with results interpretation. The UK Biobank data was conducted under application # 88976. We thank the participants in the study and the members of the research team at UK Biobank. We thank HELIX Project, Project Viva, GoDMC for providing summary statistics data.

## Author contributions

Z.Z. conceptualized and designed the study, carried out the statistical analysis, drafted the initial manuscript, and approved the final manuscript as submitted. Y.L. performed data quality control, reviewed the manuscript, and approved the final manuscript. R.J.F. conducted specimen processing, reviewed and revised the manuscript, and approved the final manuscript as submitted. J.C.C. collected the data, reviewed and revised the manuscript, and approved the final manuscript as submitted. JA.E. performed data quality control, reviewed the manuscript, and approved the final manuscript. B.H. conducted specimen processing, reviewed and revised the manuscript, and approved the final manuscript as submitted. A.H. conducted specimen processing, reviewed and revised the manuscript, and approved the final manuscript as submitted. C.A.C. conceptualized and designed the study, obtained funding, collected the data, supervised the conduct of study, critically reviewed and revised the initial manuscript, and approved the final manuscript as submitted. L.L. conceptualized and designed the study, obtained funding, collected the data, supervised the conduct of study, critically reviewed and revised the initial manuscript, and approved the final manuscript as submitted. K.H. conceptualized and designed the study, obtained funding, supervised the conduct of study, reviewed and revised the initial manuscript, and approved the final manuscript as submitted.

## Competing interests

The Authors declare the following competing interests. Z.Z. reports grants from National Institutes of Health during the conduct of the study. J.C.C. received research materials (inhaled steroids) from Merck, in order to provide medications free of cost to participants in an NIH-funded study, outside the submitted work. H.H. reports grants from National Institutes of Health and Cystic Fibrosis Foundation, personal fees from TGV-Dx and Johnson and Johnson, outside the submitted work. C.A.C. reports grants from National Institutes of Health during the conduct of the study. L.L. reports grants from National Institutes of Health during the conduct of the study. K.H. reports grants from National Institutes of Health during the conduct of the study; grants from Novartis, outside the submitted work. All other authors declare no competing interests.
