## [Peer Review File · Nature Communications]

Epigenome-wide association analysis of infant bronchiolitis severity: A multicenter prospective cohort studyREVIEWER COMMENTS

Reviewer #1 (Remarks to the Author):

Manuscript by Zhu et al describes results of the first epigenome-wide association study (EWAS) of infant bronchiolitis severity . EWAS was conducted on 625 infants hospitalized for bronchiolitis who were selected for high-quality blood DNA methylation data from the MARC-35 study. MARC-35 enrolled infants who were hospitalized with an attending physician diagnosis of bronchiolitis during three bronchiolitis seasons (November 1 to April 30) from 2011 to 2014 at 17 US sites. Differentially methylated CpGs (DMCs) for the risk of positive pressure ventilation (PPV) use, as the primary measure of bronchiolitis severity, were identified and characterized based on their association with blood immune cells, enrichment by tissue and cell types, and biological pathways. Project Viva cohort data and UK biobank data were used to investigate association of bronchiolitis severity DMCs with respiratory and immune related traits. To address causality, two-sample Mendelian randomization was performed on methylation QTLs from the GoDMC database and four respiratory traits from UK Biobank.

Main results of this investigation are 46 DMCs associated with PPV use, computationally determined to be differentially methylated in seven blood cell types, enriched in multiple tissues (such as lung) and cells (such as small airway epithelial cells, fetal lung), and biological pathways (such as T cell receptor signaling). Four DMCs were associated with asthma risk and lung function in the UK Biobank cohort, based on Mendelian Randomization analysis.

This is an important study that is well executed and reported. Some of the key strengths are uniqueness of the cohort, excellent analytical methods, and use of publicly available data to characterize the bronchiolitis severity-associated DMCs. Main weaknesses are lack of replication, lack of other datasets in this cohort (especially gene expression but also genetic data), and lack of cell specific analyses (beyond deconvolution). The results of this study add to the existing body of literature on the importance of DNA methylation in childhood respiratory diseases but do not represent a major step forward in the field.

Reviewer #2 (Remarks to the Author):

Zhu et al. present the EWAS analysis of whole blood in 625 infants (<1 year of age) hospitalised for bronchiolitis testing for association with a marker of disease severity, PPV use. Additional analyses include cell type estimation using EpiDISH and biological pathways using eFORGE and methylGSA. dmCpGs identified in the primary analysis were tested for association with respiratory and immune traits in 547 children from Project Viva – notably in DNAm data from nasal brushes, not whole blood.

Finally using GoDMC/ UK biobank data the SNPs that were associated with methylation of the bronchiolitis dmCpGs from the primary analysis were tested for association with respiratory traits including asthma and lung function in UK Biobank using Two Sample Mendelian Randomisation

The DNA extraction, DNAm measurement and processing methods followed well established methodologies.

Major comments:

1. The primary motivation of the study was to investigate “the role of the epigenome in bronchiolitis severity by applying epigenome-wide association study (EWAS) approaches to blood DNA methylation data from a multicentre prospective cohort of infants hospitalized for bronchiolitis”. However the study

design means that cause and effect relationship cannot be determined as the DNA methylation measurement is taken after hospitalisation and whether it is a consequence of disease or a cause of more severe LRTI cannot be distinguished.

2. Having calculated blood cell proportions using EpiDISH, were the proportions associated with outcome as well as the dmCpGs being associated with blood cell types?

3. Although matched gene expression data was not available in the study population, did the authors consider a look up in the EWASatlas or other datasets of match blood – RNAseq to assess the association of dmCpGs with gene expression (annotated gene or better a window around the locus)?

4. While the MR analysis provide limited support that the methylation might be on the causal pathway between LRTI in early infancy and LF growth / later asthma for a small number of differentially methylated CpGs in the primary EWAS, it is very hard to distinguish the order of effects. Is it severe LRTI – methylation – asthma or methylation – severe LRTI – asthma? Did the authors consider a lookup in cohorts with DNA methylation available at birth and lung function / asthma outcomes? E.g. the Isle of Wight Cohort (reference 32), ALSPAC or other cohorts from the PACE meta analysis conducted by Reese et al. (J Allergy Clin Immunol. 2019 Jun;143(6):2062-2074) ?

Minor comments

1. In the supplemental methods (Blood DNA Methylation Profiling and Quality Control, page 6) references should be given for the probes excluded for co-hybridization / probe SNPs if lists derived from external source, or details of the population used to identify SNPs with MAF >5% if done bespoke for this study (given GWAS data is available was this done using the studies on genotyping data?)

2. Methods line 362/363 – it should be explicitly mentioned the age of sampling in projectVIVA and tissue origin of the DNAm data.

3. EWAS summary statistics should be deposited in the EWAS catalogue

Reviewer #3 (Remarks to the Author):

This manuscript investigates the relationship between Bronchiolitis severity in infants defined by positive pressure ventilation (PPV) and DNA methylation at baseline from leukocytes. A total of 46 CpGs were differentially methylated relative to PPV. Bioinformatic analyses reveal cell-type specific signatures based on leukocytes estimated from the bulk tissue and enrichment for relevant pathways and diseases. Overall, this is an interesting study with a relatively large sample size on a difficult to reach population enhanced by causal statistical analyses. The study is limited by the measurement of bulk DNA methylation data from all leukocytes within a special hospitalized patient population. The findings are limited but their relevance is enhanced by bioinformatic analyses of top hits and the use of external data for validation and interpretation. There are several technical limitations that need to be addressed as well as clarity needed in the manuscript for the approach:

Major Limitations:

-Cell-type estimates were not adjusted for in multivariate models. This is a major limitation given the potential for immune cell-confounding. The authors adjust for surrogate variables to control for the effects of cell-types, but this is insufficient/not appropriate when actual leukocyte abundance can be estimated from the bulk DNA methylation data. The authors do perform deconvolution later, but it is unclear why these estimates are not adjusted for in the main models. I suggest that main models must adjust for cell-type estimates.

-Methods; it is unclear how the authors determine if the CpG hits are hyper or hypomethylated in the specific cell-types. Given that the DNA methylation information comes from a mixture of cell-types it is not clear if this is even possible with the estimates derived from the bulk tissue itself. This needs to be further explained as to how this was determined in terms of specific cell-types being hyper or hypomethylated at specific sites.

-Was medical treatment or treatments considered as confounders? For example, table 1 reports "Corticosteroid" use during hospitalization but this is not included as a confounder. Could treatment regimen based on disease severity be driving the associations? This is particularly plausible if treatments target inflammation affecting leukocyte distribution.

-While the single CpG analyses are interesting, regional DNA methylation analyses could enhance the biological interpretation of findings. This should be considered in the manuscript to find entire genomic regions that might relate to PPV.

Reviewer #1 (Remarks to the Author):

Manuscript by Zhu et al describes results of the first epigenome-wide association study (EWAS) of infant bronchiolitis severity. EWAS was conducted on 625 infants hospitalized for bronchiolitis who were selected for high-quality blood DNA methylation data from the MARC-35 study. MARC-35 enrolled infants who were hospitalized with an attending physician diagnosis of bronchiolitis during three bronchiolitis seasons (November 1 to April 30) from 2011 to 2014 at 17 US sites. Differentially methylated CpGs (DMCs) for the risk of positive pressure ventilation (PPV) use, as the primary measure of bronchiolitis severity, were identified and characterized based on their association with blood immune cells, enrichment by tissue and cell types, and biological pathways. Project Viva cohort data and UK biobank data were used to investigate association of bronchiolitis severity DMCs with respiratory and immune related traits. To address causality, two-sample Mendelian randomization was performed on methylation QTLs from the GoDMC database and four respiratory traits from UK Biobank.

Main results of this investigation are 46 DMCs associated with PPV use, computationally determined to be differentially methylated in seven blood cell types, enriched in multiple tissues (such as lung) and cells (such as small airway epithelial cells, fetal lung), and biological pathways (such as T cell receptor signaling). Four DMCs were associated with asthma risk and lung function in the UK Biobank cohort, based on Mendelian Randomization analysis.

This is an important study that is well executed and reported. Some of the key strengths are uniqueness of the cohort, excellent analytical methods, and use of publicly available data to characterize the bronchiolitis severity-associated DMCs. Main weaknesses are lack of replication, lack of other datasets in this cohort (especially gene expression but also genetic data), and lack of cell specific analyses (beyond deconvolution). The results of this study add to the existing body of literature on the importance of DNA methylation in childhood respiratory diseases but do not represent a major step forward in the field.

Authors' response: We thank the reviewer for the positive comments and for pointing out the limitations. In this revision, to address the limitation of the lack of paired gene expression data, we have included blood-based cis-expression quantitative trait methylation (eQTM) data from the Human Early Life Exposome (HELIX) project (Ruiz-Arenas et al. 2022). For the details of this dataset and analysis, please see the response to Reviewer #2, Major Comment #3. Additionally, as suggested, we have acknowledged these limitations in the Discussion section (page 14, paragraph 1). The text now states:

“Fourth, although we have used the cis-eQTM data from the HELIX project to investigate the association of CpGs and gene expression, the current study lacks paired transcriptome data in blood to investigate the effect of DNA methylation on gene expression. Fifth, while nearly half of the identified CpGs were associated with respiratory and immune traits in an independent study, our inferences warrant external replication using the same bronchiolitis severity outcome. However, to our best knowledge, DNA methylation data with the same outcome are not currently available.”

References:

Ruiz-Arenas C, Hernandez-Ferrer C, Vives-Usano M, Marí S, Quintela I, Mason D, Cadiou S, Casas M, Andrusaityte S, Gutzkow KB, Vafeiadi M, Wright J, Lepeule J, Grazuleviciene R, Chatzi L, Carracedo Á, Estivill X, Marti E, Escaramís G, Vrijheid M, González JR, Bustamante M. Identification of autosomal cis expression quantitative trait methylation (cis eQTM) in children's blood. *Elife*. 2022;11:e65310.

eQTM dataset URL: <https://datadryad.org/stash/dataset/doi:10.5061/dryad.fxpnvx0t0>

Reviewer #2 (Remarks to the Author):

Zhu et al. present the EWAS analysis of whole blood in 625 infants (<1 year of age) hospitalised for bronchiolitis testing for association with a marker of disease severity, PPV use. Additional analyses include cell type estimation using EpiDISH and biological pathways using eFORGE and methylGSA. dmCpGs identified in the primary analysis were tested for association with respiratory and immune traits in 547 children from Project Viva – notably in DNAm data from nasal brushes, not whole blood. Finally using GoDMC/ UK biobank data the SNPs that were associated with methylation of the bronchiolitis dmCpGs from the primary analysis were tested for association with respiratory traits including asthma and lung function in UK Biobank using Two Sample Mendelian Randomisation. The DNA extraction, DNAm measurement and processing methods followed well established methodologies.

Authors' response: We thank the reviewer for the positive comment.

Major comments:

1. The primary motivation of the study was to investigate “the role of the epigenome in bronchiolitis severity by applying epigenome-wide association study (EWAS) approaches to blood DNA methylation data from a multicentre prospective cohort of infants hospitalized for bronchiolitis”. However the study design means that cause and effect relationship cannot be determined as the DNA methylation measurement is taken after hospitalisation and whether it is a consequence of disease or a cause of more severe LRTI cannot be distinguished.

Authors' response: We thank and agree with the reviewer for the important comment. We note that the cross-sectional design limited us to investigating the exact causal link between the DNA methylation signature and bronchiolitis severity. As suggested, we have highlighted this potential limitation in the Discussion section (page 13, paragraph 2). The text states:

“First, the cross-sectional design limited us to investigate the exact causal link between the DNA methylation signature and bronchiolitis severity.”

2. Having calculated blood cell proportions using EpiDISH, were the proportions associated with outcome as well as the dmCpGs being associated with blood cell types?

Authors' response: As suggested, we have performed the association analysis of the blood cell proportions with PPV use (please see the table below).

Cell type	Estimate	SE	P	FDR
B	-0.02	0.01	0.12	0.14
NK	-0.01	0.01	0.01	0.01
CD4+T	-0.06	0.02	0.00	0.01
CD8+T	-0.02	0.01	0.06	0.08
Mono	-0.02	0.01	0.01	0.01
Neutrophil	0.13	0.03	0.00	7.80×10^{-5}
Eosinophil	-0.01	0.00	0.14	0.14

We also acknowledge that DMCs can be potentially associated with blood cell types. In our primary association model, we have used surrogate variable analysis to adjust for potential confounding effect from cell type mixtures and technical batches. In addition, we have performed a sensitivity analysis by adjusting for 7 blood cell types (B cells, NK cells, CD4T cells, CD8T cells, monocytes, neutrophils, eosinophils) from EpiDISH cell type deconvolution analysis and compared with the primary model. As shown in the cluster plot (please see below), the effect estimates for all 46 DMCs are highly consistent between the primary model (i.e., the one adjusting for surrogate variables and **without** adjusting for cell types) and the sensitivity model (i.e., the other with **adjusting for cell types**). This indicates that the surrogate variables sufficiently control for cell type mixtures and can additionally control for unknown batch effects.

3. Although matched gene expression data was not available in the study population, did the authors consider a look up in the EWASatlas or other datasets of match blood – RNAseq to assess the association of dmGpGs with gene expression (annotated gene or better a window around the locus)?

Authors' response: We agree with the reviewer that it is important to investigate the association of DMCs with gene expression in blood. Although the matched blood gene expression data were not available in our cohort, as suggested by the reviewer, we have identified blood-based cis-expression quantitative trait methylation (eQTM) data from the Human Early Life Exposome (HELIX) project (Ruiz-Arenas et al. 2022). The HELIX Project has collected blood specimens from 823 European ancestry children. The HELIX project measured blood DNA methylation and gene expression with the Illumina 450K and the Affymetrix HTA v2 arrays, respectively. The relationship between methylation levels and expression of nearby genes ($\pm 500\text{kb}$ window centered at the transcription start site) was assessed by fitting 13.6 million linear regressions adjusting for age, sex, cohort, and blood cell composition. Among the 46 DMCs from this study, we have identified 269 CpG-gene pairs from the cis-eQTM data, of which one pair showed a significant association (cg12896170 and *TRIM27*, $\log_2\text{FC}=-0.07$, $\text{FDR}=2.39\times 10^{-4}$, **Supplementary Table 5** [please see below]). Accordingly, we have added the description of cis-

eQTM dataset to the Methods section (page 19, paragraph 1) and Supplementary Methods, and summarized the results in the Results section (page 8, paragraph 2). The text now states:

Methods section: “We investigated the association of DMCs with transcription of nearby genes using publicly available blood-based cis-eQTM data from 823 European ancestry children (mean age 8 years) in the HELIX project.”

Results section: “Among the severity-related DMCs, we have identified 269 CpG-gene pairs from the cis-expression quantitative trait methylation (eQTM) data from the Human Early Life Exposome (HELIX) project, of which one pair showed a significant association (cg12896170 and *TRIM27*, $\log_2FC=-0.07$, $FDR=2.39\times 10^{-4}$; **Supplementary Table 5**).”

Additionally, we have acknowledged the lack of matched blood gene expression data in our cohort as a potential limitation in the Discussion section (page 14, paragraph 1). The text now states:

“Fourth, although we have used the cis-eQTM data from the HELIX project to investigate the association of CpGs and gene expression, the current study lacks paired transcriptome data in blood to investigate the effect of DNA methylation on gene expression.”

References:

Ruiz-Arenas C, Hernandez-Ferrer C, Vives-Usano M, Marí S, Quintela I, Mason D, Cadiou S, Casas M, Andrusaityte S, Gutzkow KB, Vafeiadi M, Wright J, Lepeule J, Grazuleviciene R, Chatzi L, Carracedo Á, Estivill X, Marti E, Escaramís G, Vrijheid M, González JR, Bustamante M. Identification of autosomal cis expression quantitative trait methylation (cis eQTMs) in children's blood. *Elife*. 2022;11:e65310.

eQTM dataset URL: <https://datadryad.org/stash/dataset/doi:10.5061/dryad.fxpnvx0t0>

Supplementary Table 5. Cis-eQTM analysis of the available DMCs in the HELIX Project

CpG	TC	CpG_chr	CpG_pos	CpG_gene	TC_gene	log2FC	SE	P-value	FDR	sigPair
cg12896170	TC06001474.hg.1	chr6	28890069	TRIM27	TRIM27	-0.07	0.14	8.89×10 ⁻⁷	2.39×10 ⁻⁴	TRUE
cg26387667	TC15001296.hg.1	chr15	45412590	DUOXAI	SPG11	0.01	0.24	0.69	0.97	FALSE
cg26387667	TC15001298.hg.1	chr15	45412590	DUOXAI		-0.01	0.69	0.93	0.97	FALSE
cg26387667	TC15001297.hg.1	chr15	45412590	DUOXAI	PATL2	-0.11	0.39	0.01	0.32	FALSE
cg26387667	TC15000342.hg.1	chr15	45412590	DUOXAI	B2M	0.01	0.13	0.64	0.97	FALSE
cg26387667	TC15002179.hg.1	chr15	45412590	DUOXAI	B2M	0.00	0.05	0.46	0.88	FALSE
cg26387667	TC15000343.hg.1	chr15	45412590	DUOXAI	TRIM69	-0.02	0.20	0.36	0.88	FALSE
cg26387667	TC15002180.hg.1	chr15	45412590	DUOXAI		-0.02	0.72	0.76	0.97	FALSE
cg26387667	TC15000344.hg.1	chr15	45412590	DUOXAI		-0.02	0.37	0.54	0.96	FALSE
cg26387667	TC15000345.hg.1	chr15	45412590	DUOXAI		-0.08	0.46	0.08	0.75	FALSE
cg26387667	TC15002517.hg.1	chr15	45412590	DUOXAI		-0.09	0.44	0.05	0.75	FALSE
cg26387667	TC15000346.hg.1	chr15	45412590	DUOXAI		0.01	0.38	0.79	0.97	FALSE
cg26387667	TC15001299.hg.1	chr15	45412590	DUOXAI		-0.07	0.36	0.05	0.75	FALSE
cg26387667	TC15001300.hg.1	chr15	45412590	DUOXAI		0.00	0.43	0.95	0.97	FALSE
cg26387667	TC15000347.hg.1	chr15	45412590	DUOXAI	C15orf43	0.03	0.25	0.20	0.84	FALSE
cg26387667	TC15000348.hg.1	chr15	45412590	DUOXAI		-0.05	0.40	0.19	0.84	FALSE
cg26387667	TC15000349.hg.1	chr15	45412590	DUOXAI		0.00	0.49	0.94	0.97	FALSE
cg26387667	TC15001301.hg.1	chr15	45412590	DUOXAI		-0.03	0.39	0.48	0.89	FALSE
cg26387667	TC15000350.hg.1	chr15	45412590	DUOXAI	SORD	-0.04	0.29	0.15	0.81	FALSE
cg26387667	TC15002181.hg.1	chr15	45412590	DUOXAI	SORD	-0.06	0.41	0.13	0.79	FALSE
cg26387667	TC15001303.hg.1	chr15	45412590	DUOXAI		-0.04	0.51	0.41	0.88	FALSE
cg26387667	TC15001302.hg.1	chr15	45412590	DUOXAI		-0.02	0.36	0.63	0.97	FALSE
cg26387667	TC15001304.hg.1	chr15	45412590	DUOXAI		-0.03	0.41	0.46	0.88	FALSE
cg26387667	TC15001305.hg.1	chr15	45412590	DUOXAI	DUOX2	0.01	0.08	0.11	0.75	FALSE
cg26387667	TC15001306.hg.1	chr15	45412590	DUOXAI	DUOXAI	-0.02	0.11	0.09	0.75	FALSE
cg26387667	TC15000352.hg.1	chr15	45412590	DUOXAI	DUOX1	0.00	0.07	0.68	0.97	FALSE
cg26387667	TC15001307.hg.1	chr15	45412590	DUOXAI		-0.01	0.22	0.68	0.97	FALSE
cg26387667	TC15001309.hg.1	chr15	45412590	DUOXAI		0.06	0.48	0.22	0.84	FALSE
cg26387667	TC15001310.hg.1	chr15	45412590	DUOXAI		0.06	0.48	0.22	0.84	FALSE

cg26387667	TC15001308.hg.1	chr15	45412590	DUOXAI	SHF	0.00	0.11	0.89	0.97	FALSE
cg26387667	TC15001311.hg.1	chr15	45412590	DUOXAI		-0.01	0.32	0.74	0.97	FALSE
cg26387667	TC15000354.hg.1	chr15	45412590	DUOXAI	SLC28A2	-0.01	0.12	0.20	0.84	FALSE
cg26387667	TC15001312.hg.1	chr15	45412590	DUOXAI		0.00	0.19	0.88	0.97	FALSE
cg26387667	TC15001313.hg.1	chr15	45412590	DUOXAI		-0.01	0.34	0.85	0.97	FALSE
cg26387667	TC15001314.hg.1	chr15	45412590	DUOXAI		0.03	0.39	0.37	0.88	FALSE
cg26387667	TC15002182.hg.1	chr15	45412590	DUOXAI	GATM	0.06	0.43	0.15	0.81	FALSE
cg26387667	TC15002814.hg.1	chr15	45412590	DUOXAI	GATM	-0.07	0.39	0.09	0.75	FALSE
cg26387667	TC15000356.hg.1	chr15	45412590	DUOXAI	SPATA5L1	0.02	0.33	0.46	0.88	FALSE
cg26387667	TC15002184.hg.1	chr15	45412590	DUOXAI	SPATA5L1	0.04	0.45	0.35	0.88	FALSE
cg26387667	TC15002815.hg.1	chr15	45412590	DUOXAI	GATM	-0.02	0.21	0.35	0.88	FALSE
cg26387667	TC15002518.hg.1	chr15	45412590	DUOXAI	GATM	0.00	0.21	0.96	0.97	FALSE
cg26387667	TC15001316.hg.1	chr15	45412590	DUOXAI		0.01	0.30	0.82	0.97	FALSE
cg26387667	TC15001317.hg.1	chr15	45412590	DUOXAI	RNU7-5P	0.01	0.44	0.80	0.97	FALSE
cg26387667	TC15002762.hg.1	chr15	45412590	DUOXAI		0.00	0.17	0.94	0.97	FALSE
cg26387667	TC15002185.hg.1	chr15	45412590	DUOXAI		0.03	0.31	0.31	0.88	FALSE
cg26387667	TC15002519.hg.1	chr15	45412590	DUOXAI		0.07	0.51	0.15	0.81	FALSE
cg26387667	TC15001319.hg.1	chr15	45412590	DUOXAI		0.05	0.65	0.41	0.88	FALSE
cg26387667	TC15000359.hg.1	chr15	45412590	DUOXAI		-0.06	0.59	0.34	0.88	FALSE
cg26387667	TC15002186.hg.1	chr15	45412590	DUOXAI	HMGN2P46	0.02	0.21	0.35	0.88	FALSE
cg26387667	TC15002763.hg.1	chr15	45412590	DUOXAI	HMGN2P46	0.00	0.18	0.86	0.97	FALSE
cg26387667	TC15001318.hg.1	chr15	45412590	DUOXAI	SLC30A4	-0.03	0.23	0.20	0.84	FALSE
cg26387667	TC15000360.hg.1	chr15	45412590	DUOXAI		0.04	0.41	0.38	0.88	FALSE
cg26387667	TC15000361.hg.1	chr15	45412590	DUOXAI		0.04	0.41	0.31	0.88	FALSE
cg26387667	TC15000362.hg.1	chr15	45412590	DUOXAI	BLOC1S6	0.03	0.28	0.25	0.85	FALSE
cg26387667	TC15000363.hg.1	chr15	45412590	DUOXAI		0.05	0.33	0.10	0.75	FALSE
cg09432792	TC16001128.hg.1	chr16	56352311	GNAOI	CESI	0.10	5.89	0.86	0.97	FALSE
cg09432792	TC16001129.hg.1	chr16	56352311	GNAOI	CES5A	0.01	0.59	0.80	0.97	FALSE
cg09432792	TC16001905.hg.1	chr16	56352311	GNAOI		-0.39	1.75	0.02	0.66	FALSE
cg09432792	TC16001573.hg.1	chr16	56352311	GNAOI		-0.06	1.33	0.64	0.97	FALSE
cg09432792	TC16001907.hg.1	chr16	56352311	GNAOI		0.17	2.05	0.41	0.88	FALSE
cg09432792	TC16001130.hg.1	chr16	56352311	GNAOI	LOC283856	-0.08	1.12	0.46	0.88	FALSE

cg09432792	TC16001906.hg.1	chr16	56352311	GNAOI		-0.07	0.65	0.28	0.87	FALSE
cg09432792	TC16000461.hg.1	chr16	56352311	GNAOI	GNAOI	0.01	1.23	0.92	0.97	FALSE
cg09432792	TC16001908.hg.1	chr16	56352311	GNAOI	DKFZP434H168	-0.11	1.44	0.46	0.88	FALSE
cg09432792	TC16001131.hg.1	chr16	56352311	GNAOI	DKFZP434H168	0.02	1.50	0.90	0.97	FALSE
cg09432792	TC16001574.hg.1	chr16	56352311	GNAOI	GNAOI	-0.11	1.95	0.57	0.97	FALSE
cg09432792	TC16001132.hg.1	chr16	56352311	GNAOI	AMFR	0.14	2.03	0.48	0.90	FALSE
cg09432792	TC16001133.hg.1	chr16	56352311	GNAOI	NUDT21	-0.01	1.74	0.94	0.97	FALSE
cg09432792	TC16000463.hg.1	chr16	56352311	GNAOI	OGFOD1	0.02	1.72	0.91	0.97	FALSE
cg09432792	TC16001134.hg.1	chr16	56352311	GNAOI	BBS2	-0.23	2.51	0.36	0.88	FALSE
cg09432792	TC16000464.hg.1	chr16	56352311	GNAOI	MT4	-0.06	1.73	0.75	0.97	FALSE
cg09432792	TC16000465.hg.1	chr16	56352311	GNAOI	MT3	-0.01	0.91	0.87	0.97	FALSE
cg09432792	TC16002034.hg.1	chr16	56352311	GNAOI	MT2A	0.48	6.39	0.45	0.88	FALSE
cg09432792	TC16001575.hg.1	chr16	56352311	GNAOI	MT1L	0.24	3.20	0.45	0.88	FALSE
cg09432792	TC16002075.hg.1	chr16	56352311	GNAOI	MT1L	-0.03	3.19	0.94	0.97	FALSE
cg09432792	TC16000468.hg.1	chr16	56352311	GNAOI	MT1E	0.50	2.84	0.08	0.75	FALSE
cg09432792	TC16002074.hg.1	chr16	56352311	GNAOI	MT1M	0.36	1.83	0.05	0.75	FALSE
cg09432792	TC16000469.hg.1	chr16	56352311	GNAOI	MT1JP	0.30	3.00	0.32	0.88	FALSE
cg09432792	TC16001576.hg.1	chr16	56352311	GNAOI	MT1JP	0.01	2.60	0.98	0.98	FALSE
cg09432792	TC16002035.hg.1	chr16	56352311	GNAOI	MT1A	-0.07	2.38	0.78	0.97	FALSE
cg09432792	TC16001577.hg.1	chr16	56352311	GNAOI		0.21	1.45	0.16	0.81	FALSE
cg09432792	TC16000470.hg.1	chr16	56352311	GNAOI	MT1DP	0.38	1.98	0.05	0.75	FALSE
cg09432792	TC16000471.hg.1	chr16	56352311	GNAOI		0.57	3.45	0.10	0.75	FALSE
cg09432792	TC16000472.hg.1	chr16	56352311	GNAOI	MT1B	0.20	2.49	0.42	0.88	FALSE
cg09432792	TC16000473.hg.1	chr16	56352311	GNAOI	MT1F	-0.37	4.63	0.42	0.88	FALSE
cg09432792	TC16001135.hg.1	chr16	56352311	GNAOI	MT1G	0.03	2.62	0.92	0.97	FALSE
cg09432792	TC16000474.hg.1	chr16	56352311	GNAOI	MT1H	0.07	3.01	0.81	0.97	FALSE
cg09432792	TC16000475.hg.1	chr16	56352311	GNAOI	MT1IP	-0.02	1.35	0.89	0.97	FALSE
cg09432792	TC16001578.hg.1	chr16	56352311	GNAOI	MT1IP	-0.02	1.42	0.87	0.97	FALSE
cg09432792	TC16000476.hg.1	chr16	56352311	GNAOI	MT1X	-0.14	2.88	0.63	0.97	FALSE
cg09432792	TC16000477.hg.1	chr16	56352311	GNAOI	NUP93	0.05	1.39	0.71	0.97	FALSE
cg09541576	TC02001796.hg.1	chr2	44873248	C2orf34		0.05	0.51	0.36	0.88	FALSE
cg09541576	TC02000274.hg.1	chr2	44873248	C2orf34	PPM1B	-0.02	0.19	0.22	0.84	FALSE

cg09541576	TC02001797.hg.1	chr2	44873248	C2orf34		0.02	0.38	0.53	0.96	FALSE
cg09541576	TC02004203.hg.1	chr2	44873248	C2orf34		0.00	0.28	0.91	0.97	FALSE
cg09541576	TC02000275.hg.1	chr2	44873248	C2orf34	SLC3A1	-0.04	0.12	0.00	0.24	FALSE
cg09541576	TC02001798.hg.1	chr2	44873248	C2orf34	PREPL	0.04	0.23	0.10	0.75	FALSE
cg09541576	TC02000276.hg.1	chr2	44873248	C2orf34	CAMKMT	0.00	0.19	0.95	0.97	FALSE
cg09541576	TC02003196.hg.1	chr2	44873248	C2orf34		0.01	0.73	0.91	0.97	FALSE
cg09541576	TC02001801.hg.1	chr2	44873248	C2orf34		0.05	0.43	0.22	0.84	FALSE
cg09541576	TC02004204.hg.1	chr2	44873248	C2orf34		0.02	0.22	0.46	0.88	FALSE
cg09541576	TC02003197.hg.1	chr2	44873248	C2orf34		0.05	0.38	0.23	0.84	FALSE
cg09541576	TC02004205.hg.1	chr2	44873248	C2orf34		0.02	0.47	0.67	0.97	FALSE
cg09541576	TC02001802.hg.1	chr2	44873248	C2orf34	LOC100130502	-0.01	0.12	0.35	0.88	FALSE
cg09541576	TC02004206.hg.1	chr2	44873248	C2orf34		-0.01	0.16	0.43	0.88	FALSE
cg09541576	TC02000277.hg.1	chr2	44873248	C2orf34	SIX3	-0.01	0.23	0.78	0.97	FALSE
cg09541576	TC02001803.hg.1	chr2	44873248	C2orf34	SIX3-AS1	0.02	0.28	0.50	0.91	FALSE
cg09541576	TC02004207.hg.1	chr2	44873248	C2orf34		0.04	0.26	0.16	0.81	FALSE
cg09541576	TC02004208.hg.1	chr2	44873248	C2orf34	SIX3	-0.06	0.25	0.02	0.66	FALSE
cg09541576	TC02000278.hg.1	chr2	44873248	C2orf34		0.02	0.28	0.40	0.88	FALSE
cg09541576	TC02003198.hg.1	chr2	44873248	C2orf34		0.04	0.26	0.10	0.75	FALSE
cg09541576	TC02003199.hg.1	chr2	44873248	C2orf34		0.04	0.39	0.31	0.88	FALSE
cg09541576	TC02001804.hg.1	chr2	44873248	C2orf34	SIX2	-0.03	0.34	0.34	0.88	FALSE
cg09541576	TC02001805.hg.1	chr2	44873248	C2orf34		0.01	0.23	0.57	0.97	FALSE
cg09541576	TC02004209.hg.1	chr2	44873248	C2orf34		0.01	0.23	0.56	0.97	FALSE
cg09541576	TC02003200.hg.1	chr2	44873248	C2orf34		-0.06	0.35	0.07	0.75	FALSE
cg09412707	TC04001091.hg.1	chr4	26085653			0.04	0.25	0.09	0.75	FALSE
cg09412707	TC04000177.hg.1	chr4	26085653			0.02	0.32	0.64	0.97	FALSE
cg09412707	TC04001943.hg.1	chr4	26085653			-0.02	0.30	0.46	0.88	FALSE
cg09412707	TC04000178.hg.1	chr4	26085653		SLC34A2	0.00	0.09	0.76	0.97	FALSE
cg09412707	TC04000179.hg.1	chr4	26085653			-0.03	0.33	0.34	0.88	FALSE
cg09412707	TC04000180.hg.1	chr4	26085653		SMIM20	0.01	0.21	0.71	0.97	FALSE
cg09412707	TC04001092.hg.1	chr4	26085653		SELIL3	-0.17	0.58	0.00	0.24	FALSE
cg09412707	TC04001093.hg.1	chr4	26085653			-0.03	0.25	0.20	0.84	FALSE
cg09412707	TC04002497.hg.1	chr4	26085653			-0.03	0.19	0.08	0.75	FALSE

cg09412707	TC04001944.hg.1	chr4	26085653			-0.01	0.27	0.69	0.97	FALSE
cg09412707	TC04000181.hg.1	chr4	26085653		RBPJ	0.01	0.19	0.77	0.97	FALSE
cg09412707	TC04002498.hg.1	chr4	26085653			0.00	0.27	0.92	0.97	FALSE
cg09412707	TC04001945.hg.1	chr4	26085653		RBPJ	0.06	0.33	0.06	0.75	FALSE
cg09412707	TC04001946.hg.1	chr4	26085653		RBPJ	0.05	0.44	0.26	0.87	FALSE
cg09412707	TC04001095.hg.1	chr4	26085653		CCKAR	0.00	0.16	0.75	0.97	FALSE
cg09412707	TC04000182.hg.1	chr4	26085653		TBC1D19	-0.01	0.17	0.42	0.88	FALSE
cg12214366	TC04000443.hg.1	chr4	80977132	ANTXR2		0.00	0.34	0.92	0.97	FALSE
cg12214366	TC04002609.hg.1	chr4	80977132	ANTXR2		0.01	0.30	0.72	0.97	FALSE
cg12214366	TC04000444.hg.1	chr4	80977132	ANTXR2		-0.03	0.25	0.25	0.86	FALSE
cg12214366	TC04002087.hg.1	chr4	80977132	ANTXR2		0.00	0.28	0.86	0.97	FALSE
cg12214366	TC04000445.hg.1	chr4	80977132	ANTXR2	PCAT4	-0.03	0.25	0.25	0.86	FALSE
cg12214366	TC04002088.hg.1	chr4	80977132	ANTXR2		-0.03	0.33	0.44	0.88	FALSE
cg12214366	TC04000446.hg.1	chr4	80977132	ANTXR2		0.02	0.40	0.70	0.97	FALSE
cg12214366	TC04001328.hg.1	chr4	80977132	ANTXR2	ANTXR2	0.00	0.28	0.91	0.97	FALSE
cg12214366	TC04000447.hg.1	chr4	80977132	ANTXR2	PRDM8	-0.03	0.18	0.07	0.75	FALSE
cg12214366	TC04001329.hg.1	chr4	80977132	ANTXR2		-0.02	0.22	0.27	0.87	FALSE
cg12214366	TC04002610.hg.1	chr4	80977132	ANTXR2		0.01	0.28	0.83	0.97	FALSE
cg12214366	TC04002936.hg.1	chr4	80977132	ANTXR2	FGF5	-0.02	0.12	0.10	0.75	FALSE
cg12214366	TC04002937.hg.1	chr4	80977132	ANTXR2	C4orf22	-0.01	0.12	0.42	0.88	FALSE
cg15848159	TC04001353.hg.1	chr4	85791643	WDFY3		0.00	0.19	0.91	0.97	FALSE
cg15848159	TC04002619.hg.1	chr4	85791643	WDFY3		0.00	0.19	0.86	0.97	FALSE
cg15848159	TC04001354.hg.1	chr4	85791643	WDFY3	NKX6-1	0.01	0.09	0.36	0.88	FALSE
cg15848159	TC04000462.hg.1	chr4	85791643	WDFY3	CDS1	-0.01	0.08	0.23	0.84	FALSE
cg15848159	TC04000463.hg.1	chr4	85791643	WDFY3		0.00	0.24	0.85	0.97	FALSE
cg15848159	TC04001356.hg.1	chr4	85791643	WDFY3		-0.04	0.24	0.07	0.75	FALSE
cg15848159	TC04000464.hg.1	chr4	85791643	WDFY3	WDFY3-AS2	-0.02	0.10	0.02	0.66	FALSE
cg15848159	TC04002091.hg.1	chr4	85791643	WDFY3	WDFY3-AS2	-0.01	0.10	0.30	0.88	FALSE
cg15848159	TC04001355.hg.1	chr4	85791643	WDFY3	WDFY3	0.00	0.21	0.88	0.97	FALSE
cg15848159	TC04000465.hg.1	chr4	85791643	WDFY3		-0.02	0.19	0.24	0.84	FALSE
cg15848159	TC04002092.hg.1	chr4	85791643	WDFY3		0.01	0.17	0.74	0.97	FALSE
cg12547959	TC05000082.hg.1	chr5	14326153	TRIO		0.01	0.17	0.73	0.97	FALSE

cg12547959	TC05001171.hg.1	chr5	14326153	TRIO	DNAH5	0.00	0.03	0.66	0.97	FALSE
cg12547959	TC05002282.hg.1	chr5	14326153	TRIO		0.01	0.14	0.69	0.97	FALSE
cg12547959	TC05000083.hg.1	chr5	14326153	TRIO	TRIO	-0.02	0.10	0.11	0.75	FALSE
cg12547959	TC05002283.hg.1	chr5	14326153	TRIO	TRIO	-0.01	0.16	0.66	0.97	FALSE
cg12547959	TC05000084.hg.1	chr5	14326153	TRIO	FAM105A	0.00	0.17	0.89	0.97	FALSE
cg12547959	TC05000085.hg.1	chr5	14326153	TRIO		0.01	0.23	0.62	0.97	FALSE
cg12547959	TC05002284.hg.1	chr5	14326153	TRIO		0.02	0.25	0.46	0.88	FALSE
cg12547959	TC05002907.hg.1	chr5	14326153	TRIO		0.00	0.13	0.98	0.98	FALSE
cg12547959	TC05000086.hg.1	chr5	14326153	TRIO	OTULIN	-0.01	0.09	0.16	0.81	FALSE
cg12896170	TC06001440.hg.1	chr6	28890069	TRIM27	ZSCAN23	-0.01	0.16	0.60	0.97	FALSE
cg12896170	TC06000285.hg.1	chr6	28890069	TRIM27	COX11P1	-0.04	0.26	0.09	0.75	FALSE
cg12896170	TC06001442.hg.1	chr6	28890069	TRIM27		-0.03	0.18	0.13	0.81	FALSE
cg12896170	TC06000286.hg.1	chr6	28890069	TRIM27	GPX5	0.01	0.11	0.24	0.84	FALSE
cg12896170	TC06001444.hg.1	chr6	28890069	TRIM27	GPX6	0.00	0.12	0.71	0.97	FALSE
cg12896170	TC06000288.hg.1	chr6	28890069	TRIM27		0.00	0.16	0.81	0.97	FALSE
cg12896170	TC06002673.hg.1	chr6	28890069	TRIM27		-0.01	0.17	0.37	0.88	FALSE
cg12896170	TC06001447.hg.1	chr6	28890069	TRIM27		-0.06	0.22	0.01	0.32	FALSE
cg12896170	TC06002674.hg.1	chr6	28890069	TRIM27		0.04	0.20	0.04	0.75	FALSE
cg12896170	TC06001446.hg.1	chr6	28890069	TRIM27	ZBED9	-0.01	0.09	0.22	0.84	FALSE
cg12896170	TC06000290.hg.1	chr6	28890069	TRIM27		0.00	0.14	0.88	0.97	FALSE
cg12896170	TC06001450.hg.1	chr6	28890069	TRIM27		0.03	0.18	0.14	0.81	FALSE
cg12896170	TC06003558.hg.1	chr6	28890069	TRIM27		0.01	0.17	0.53	0.96	FALSE
cg12896170	TC06001451.hg.1	chr6	28890069	TRIM27		0.00	0.12	0.94	0.97	FALSE
cg12896170	TC06000291.hg.1	chr6	28890069	TRIM27		-0.01	0.12	0.38	0.88	FALSE
cg12896170	TC06000292.hg.1	chr6	28890069	TRIM27		0.01	0.10	0.24	0.84	FALSE
cg12896170	TC06000293.hg.1	chr6	28890069	TRIM27		-0.02	0.18	0.35	0.88	FALSE
cg12896170	TC06001455.hg.1	chr6	28890069	TRIM27		-0.03	0.23	0.16	0.81	FALSE
cg12896170	TC06001456.hg.1	chr6	28890069	TRIM27		-0.01	0.18	0.67	0.97	FALSE
cg12896170	TC06000295.hg.1	chr6	28890069	TRIM27		-0.03	0.22	0.19	0.84	FALSE
cg12896170	TC06003559.hg.1	chr6	28890069	TRIM27		0.02	0.18	0.38	0.88	FALSE
cg12896170	TC06001458.hg.1	chr6	28890069	TRIM27		0.02	0.18	0.26	0.87	FALSE
cg12896170	TC06001459.hg.1	chr6	28890069	TRIM27		-0.02	0.20	0.41	0.88	FALSE

cg12896170	TC06000296.hg.1	chr6	28890069	TRIM27		0.01	0.23	0.68	0.97	FALSE
cg12896170	TC06000297.hg.1	chr6	28890069	TRIM27		0.00	0.16	0.78	0.97	FALSE
cg12896170	TC06001460.hg.1	chr6	28890069	TRIM27		0.02	0.16	0.13	0.79	FALSE
cg12896170	TC06003560.hg.1	chr6	28890069	TRIM27		-0.02	0.20	0.22	0.84	FALSE
cg12896170	TC06001461.hg.1	chr6	28890069	TRIM27		-0.02	0.15	0.28	0.88	FALSE
cg12896170	TC06001462.hg.1	chr6	28890069	TRIM27		0.03	0.24	0.21	0.84	FALSE
cg12896170	TC06001463.hg.1	chr6	28890069	TRIM27		0.00	0.12	0.78	0.97	FALSE
cg12896170	TC06001464.hg.1	chr6	28890069	TRIM27		0.00	0.12	0.91	0.97	FALSE
cg12896170	TC06001465.hg.1	chr6	28890069	TRIM27		-0.01	0.21	0.55	0.97	FALSE
cg12896170	TC06001466.hg.1	chr6	28890069	TRIM27		0.01	0.15	0.43	0.88	FALSE
cg12896170	TC06001467.hg.1	chr6	28890069	TRIM27		-0.01	0.17	0.44	0.88	FALSE
cg12896170	TC06001470.hg.1	chr6	28890069	TRIM27		0.01	0.11	0.60	0.97	FALSE
cg12896170	TC06001469.hg.1	chr6	28890069	TRIM27		-0.02	0.18	0.39	0.88	FALSE
cg12896170	TC06003561.hg.1	chr6	28890069	TRIM27		0.01	0.16	0.74	0.97	FALSE
cg12896170	TC06001471.hg.1	chr6	28890069	TRIM27	LINC01623	0.00	0.26	0.98	0.98	FALSE
cg12896170	TC06003562.hg.1	chr6	28890069	TRIM27		-0.02	0.22	0.47	0.88	FALSE
cg12896170	TC06001472.hg.1	chr6	28890069	TRIM27		0.01	0.11	0.60	0.97	FALSE
cg12896170	TC06001473.hg.1	chr6	28890069	TRIM27		0.00	0.15	0.74	0.97	FALSE
cg12896170	TC06000299.hg.1	chr6	28890069	TRIM27	HCG14	0.01	0.19	0.46	0.88	FALSE
cg12896170	TC06002675.hg.1	chr6	28890069	TRIM27		0.00	0.19	0.86	0.97	FALSE
cg12896170	TC06000300.hg.1	chr6	28890069	TRIM27		0.01	0.21	0.79	0.97	FALSE
cg12896170	TC06000301.hg.1	chr6	28890069	TRIM27		-0.02	0.19	0.40	0.88	FALSE
cg12896170	TC06001475.hg.1	chr6	28890069	TRIM27		0.03	0.16	0.08	0.75	FALSE
cg12896170	TC06001476.hg.1	chr6	28890069	TRIM27		-0.02	0.23	0.35	0.88	FALSE
cg12896170	TC06000302.hg.1	chr6	28890069	TRIM27		0.00	0.14	0.93	0.97	FALSE
cg12896170	TC06000303.hg.1	chr6	28890069	TRIM27		-0.01	0.13	0.28	0.87	FALSE
cg12896170	TC06003563.hg.1	chr6	28890069	TRIM27		-0.03	0.24	0.24	0.84	FALSE
cg12896170	TC06000304.hg.1	chr6	28890069	TRIM27		-0.04	0.19	0.02	0.66	FALSE
cg12896170	TC06001477.hg.1	chr6	28890069	TRIM27		-0.01	0.13	0.28	0.87	FALSE
cg12896170	TC06001478.hg.1	chr6	28890069	TRIM27		0.03	0.24	0.21	0.84	FALSE
cg12896170	TC06000305.hg.1	chr6	28890069	TRIM27		-0.02	0.18	0.41	0.88	FALSE
cg12896170	TC06002676.hg.1	chr6	28890069	TRIM27		0.01	0.09	0.16	0.81	FALSE

cg12896170	TC06000307.hg.1	chr6	28890069	TRIM27		-0.03	0.20	0.14	0.81	FALSE
cg12896170	TC06001479.hg.1	chr6	28890069	TRIM27	ZNF311	-0.01	0.09	0.20	0.84	FALSE
cg12896170	TC06003564.hg.1	chr6	28890069	TRIM27		0.00	0.19	0.96	0.97	FALSE
cg12896170	TC06000308.hg.1	chr6	28890069	TRIM27	LOC100129636	0.01	0.16	0.67	0.97	FALSE
cg12896170	TC06001480.hg.1	chr6	28890069	TRIM27	OR2WI	0.01	0.25	0.68	0.97	FALSE
cg12896170	TC06001481.hg.1	chr6	28890069	TRIM27	OR2B3	0.03	0.21	0.11	0.75	FALSE
cg12896170	TC06002678.hg.1	chr6	28890069	TRIM27		-0.01	0.19	0.66	0.97	FALSE
cg12896170	TC06000309.hg.1	chr6	28890069	TRIM27	OR2J1	0.00	0.20	0.94	0.97	FALSE
cg12896170	TC06000310.hg.1	chr6	28890069	TRIM27	OR2J3	-0.02	0.26	0.55	0.97	FALSE
cg12896170	TC06000311.hg.1	chr6	28890069	TRIM27		0.01	0.20	0.61	0.97	FALSE
cg12896170	TC06002679.hg.1	chr6	28890069	TRIM27		0.00	0.22	0.88	0.97	FALSE
cg12896170	TC06003565.hg.1	chr6	28890069	TRIM27		0.03	0.16	0.08	0.75	FALSE
cg12896170	TC06000312.hg.1	chr6	28890069	TRIM27	OR2J2	0.00	0.22	0.87	0.97	FALSE
cg12896170	TC06002681.hg.1	chr6	28890069	TRIM27		0.02	0.21	0.24	0.84	FALSE
cg12896170	TC06000313.hg.1	chr6	28890069	TRIM27		0.02	0.22	0.47	0.89	FALSE
cg12896170	TC06002682.hg.1	chr6	28890069	TRIM27		0.00	0.22	0.94	0.97	FALSE
cg12896170	TC06000314.hg.1	chr6	28890069	TRIM27	OR14J1	0.01	0.23	0.76	0.97	FALSE
cg12896170	TC06000315.hg.1	chr6	28890069	TRIM27	OR12D2	-0.01	0.19	0.66	0.97	FALSE
cg12896170	TC06004147.hg.1	chr6	28890069	TRIM27	OR5V1	0.02	0.26	0.37	0.88	FALSE
cg24800630	TC06003144.hg.1	chr6	1.57E+08			0.01	0.38	0.85	0.97	FALSE
cg24800630	TC06003145.hg.1	chr6	1.57E+08			0.00	0.25	0.87	0.97	FALSE
cg24800630	TC06003146.hg.1	chr6	1.57E+08			0.12	0.79	0.11	0.75	FALSE
cg24800630	TC06003942.hg.1	chr6	1.57E+08			0.02	0.42	0.72	0.97	FALSE
cg24800630	TC06001122.hg.1	chr6	1.57E+08			-0.01	0.53	0.92	0.97	FALSE
cg24800630	TC06002252.hg.1	chr6	1.57E+08			0.01	0.28	0.67	0.97	FALSE
cg24800630	TC06003943.hg.1	chr6	1.57E+08			0.01	0.27	0.65	0.97	FALSE
cg24800630	TC06003944.hg.1	chr6	1.57E+08			0.01	0.25	0.59	0.97	FALSE
cg24800630	TC06002253.hg.1	chr6	1.57E+08			0.03	0.45	0.55	0.97	FALSE
cg24800630	TC06001123.hg.1	chr6	1.57E+08		ARID1B	0.01	0.21	0.61	0.97	FALSE
cg24800630	TC06003148.hg.1	chr6	1.57E+08		ARID1B	-0.03	0.33	0.40	0.88	FALSE
cg04089246	TC07001361.hg.1	chr7	47579217	TNS3		-0.02	0.25	0.41	0.88	FALSE
cg04089246	TC07002930.hg.1	chr7	47579217	TNS3		-0.03	0.21	0.11	0.75	FALSE

cg04089246	TC07001362.hg.1	chr7	47579217	TNS3		-0.06	0.27	0.02	0.66	FALSE
cg04089246	TC07002931.hg.1	chr7	47579217	TNS3		-0.01	0.26	0.67	0.97	FALSE
cg04089246	TC07001363.hg.1	chr7	47579217	TNS3	TNS3	0.03	0.21	0.11	0.75	FALSE
cg04089246	TC07002301.hg.1	chr7	47579217	TNS3		-0.04	0.31	0.21	0.84	FALSE
cg04089246	TC07000302.hg.1	chr7	47579217	TNS3	LINC01447	0.00	0.20	0.95	0.97	FALSE
cg04089246	TC07002302.hg.1	chr7	47579217	TNS3		-0.01	0.21	0.59	0.97	FALSE
cg04089246	TC07000303.hg.1	chr7	47579217	TNS3	C7orf65	0.01	0.30	0.78	0.97	FALSE
cg04089246	TC07000304.hg.1	chr7	47579217	TNS3	LINC00525	0.00	0.33	0.88	0.97	FALSE
cg04089246	TC07002303.hg.1	chr7	47579217	TNS3		-0.07	0.33	0.03	0.75	FALSE
cg04089246	TC07000305.hg.1	chr7	47579217	TNS3	C7orf69	-0.01	0.21	0.53	0.96	FALSE
cg04089246	TC07001365.hg.1	chr7	47579217	TNS3	PKD1L1	0.00	0.07	0.54	0.96	FALSE
cg04089246	TC07002932.hg.1	chr7	47579217	TNS3		0.03	0.45	0.44	0.88	FALSE
cg04089246	TC07001364.hg.1	chr7	47579217	TNS3	HUS1	0.01	0.21	0.65	0.97	FALSE
cg04089246	TC07002933.hg.1	chr7	47579217	TNS3	HUS1	0.02	0.33	0.63	0.97	FALSE
cg04089246	TC07001366.hg.1	chr7	47579217	TNS3	SUN3	0.03	0.12	0.04	0.75	FALSE
cg04089246	TC07000306.hg.1	chr7	47579217	TNS3	C7orf57	0.00	0.13	0.88	0.97	FALSE

4. While the MR analysis provide limited support that the methylation might be on the causal pathway between LRTI in early infancy and LF growth / later asthma for a small number of differentially methylated CpGs in the primary EWAS, it is very hard to distinguish the order of effects. Is it severe LRTI – methylation – asthma or methylation – severe LRTI – asthma? Did the authors consider a lookup in cohorts with DNA methylation available at birth and lung function / asthma outcomes? E.g. the Isle of Wight Cohort (reference 32), ALSPAC or other cohorts from the PACE meta analysis conducted by Reese et al. (*J Allergy Clin Immunol.* 2019 Jun;143(6):2062-2074)?

Authors' response: We thank the reviewer for the important comment. We agree that it is hard to distinguish the order of effects between LRTI and methylation in early infancy on lung function growth or asthma development in childhood. As suggested, we have performed a look up of our 46 DMCs in PACE meta-analysis and Isle of Wight Cohort. We have compared our DMCs with Table E2 of Reese et al. *JACI* 2019 (PACE meta-analysis) and Table S2 of Mukherjee et al. *ERJ* 2021 (Isle of Wight Cohort). Unfortunately, we did not find any overlap between our DMCs and CpGs in these studies. In addition, we have also acknowledged the importance of investigating the relationship of DNA methylation at birth/early infancy with respiratory outcomes in childhood in the Discussion section (pages 13-14, paragraph 2). The text now states:

“Second, although our Mendelian randomization analysis showed the association of severity-related DMCs in infancy with respiratory outcomes in later life (e.g., asthma and lung function), it is important to investigate the association of these CpGs in infancy with respiratory outcomes in later life in a longitudinal design (Reese et al. 2019 and Mukherjee et al. 2021).”

References:

Reese SE, Xu CJ, den Dekker HT, Lee MK, Sikdar S, Ruiz-Arenas C, et al. Epigenome-wide meta-analysis of DNA methylation and childhood asthma. *J Allergy Clin Immunol.* 2019;143(6):2062-2074.

Mukherjee N, Arathimos R, Chen S, Kheirkhah Rahimabad P, Han L, Zhang H, Holloway JW, Relton C, Henderson AJ, Arshad SH, Ewart S, Karmaus W. DNA methylation at birth is associated with lung function development until age 26 years. *Eur Respir J.* 2021;57(4):2003505.

Minor comments

1. In the supplemental methods (Blood DNA Methylation Profiling and Quality Control, page 6) references should be given for the probes excluded for co-hybridization / probe SNPs if lists derived from external source, or details of the population used to identify SNPs with MAF >5% if done bespoke for this study (given GWAS data is available was this done using the studies on genotyping data?)

Authors' response: We thank the reviewer for the comment. We have used an external source (Pidsley et al. 2016) to exclude probes for co-hybridization / probe SNPs. Specifically, we used the list from Tables S1, S4, S5, and S6 of the Pidsley et al. study to exclude probes in the current study. We have added this paper to the reference (reference #5 in the Online Supplement).

Reference:

Pidsley R, Zotenko E, Peters TJ, Lawrence MG, Risbridger GP, Molloy P, Van Djik S, Muhlhausler B, Stirzaker C, Clark SJ. Critical evaluation of the Illumina MethylationEPIC BeadChip microarray for whole-genome DNA methylation profiling. *Genome Biol.* 2016;17(1):208.

2. Methods line 362/363 – it should be explicitly mentioned the age of sampling in projectVIVA and tissue origin of the DNAm data.

Authors' response: As suggested by the reviewer, we have added the description of the Project Viva's age of sampling and specimen type to the Methods section (page 19, paragraph 2). The text now states:

“The Project Viva study collected nasal swabs from the anterior nares of 547 children (mean age 12.9 years) and measured DNA methylation with the Infinium MethylationEPIC BeadChip.”

3. EWAS summary statistics should be deposited in the EWAS catalogue

Authors' response: We have contacted the EWAS catalogue (ewascatalog@outlook.com). However, the email address gave an automatic reply indicating that the person who was working on EWAS summary statistics deposition is no longer employed by the University of Bristol. Thus, we have deposited the EWAS summary statistics at our research website <http://lianglab.rc.fas.harvard.edu/BronchiolitisSeverityEWAS/>. We have also updated the Data Availability statement.

Reviewer #3 (Remarks to the Author):

This manuscript investigates the relationship between Bronchiolitis severity in infants defined by positive pressure ventilation (PPV) and DNA methylation at baseline from leukocytes. A total of 46 CpGs were differentially methylated relative to PPV. Bioinformatic analyses reveal cell-type specific signatures based on leukocytes estimated from the bulk tissue and enrichment for relevant pathways and diseases. Overall, this is an interesting study with a relatively large sample size on a difficult to reach population enhanced by causal statistical analyses. The study is limited by the measurement of bulk DNA methylation data from all leukocytes within a special

hospitalized patient population. The findings are limited but their relevance is enhanced by bioinformatic analyses of top hits and the use of external data for validation and interpretation. There are several technical limitations that need to be addressed as well as clarity needed in the manuscript for the approach:

Authors' response: We thank the reviewer for the positive comment and for pointing out the limitations.

Major Limitations:

1. Cell-type estimates were not adjusted for in multivariate models. This is a major limitation given the potential for immune cell-confounding. The authors adjust for surrogate variables to control for the effects of cell-types, but this is insufficient/not appropriate when actual leukocyte abundance can be estimated from the bulk DNA methylation data. The authors do perform deconvolution later, but it is unclear why these estimates are not adjusted for in the main models. I suggest that main models must adjust for cell-type estimates.

Authors' response: We appreciate the opportunity to clarify this important point. There were two main reasons why we did not adjust for cell type in the primary model. First, the complete blood count (CBC) from the clinical data has missingness. Thus, we did not adjust for CBCs as covariates to maintain full EWAS sample size. Second, we have performed a sensitivity analysis by adjusting for 7 blood cell types (B cells, NK cells, CD4T cells, CD8T cells, monocytes, neutrophils, and eosinophils) from EpiDISH cell type deconvolution analysis and compared with the primary model. As shown in the cluster plot (please see below), the effect estimates for all 46 DMCs are highly consistent between the primary model (i.e., the one adjusting for surrogate variables and **without** adjusting for cell types) and the sensitivity model (i.e., the other with **adjusting for cell types**). This indicates that the surrogate variables sufficiently control for cell type mixtures and can additionally control for unknown batch effects (i.e., without adjusting cell types did not affect the primary association results).

2. Methods; it is unclear how the authors determine if the CpG hits are hyper or hypomethylated in the specific cell-types. Given that the DNA methylation information comes from a mixture of cell-types it is not clear if this is even possible with the estimates derived from the bulk tissue itself. This needs to be further explained as to how this was determined in terms of specific cell-types being hyper or hypomethylated at specific sites.

Authors' response: We thank the reviewer for the comment. We followed the standard procedure in *EpiDISH* R package (Zheng et al. 2018). The *EpiDISH* has two main steps: 1) it infers the proportions of *a priori* known cell-types present in a sample representing a mixture of such cell-types. 2) CellDMC function allows the identification of differentially methylated cell types and their directionality of change in EWAS of a specific outcome (i.e., PPV use in this study). Thus, the step 2 allows the identification of the hyper or hypomethylation of specific CpGs in each cell type.

References:

Zheng SC, Breeze CE, Beck S, Teschendorff AE. Identification of differentially methylated cell types in epigenome-wide association studies. *Nat Methods*. 2018;15(12):1059-1066.

EpiDISH R package URL: <https://bioconductor.org/packages/release/bioc/html/EpiDISH.html>

3. Was medical treatment or treatments considered as confounders? For example, table 1 reports “Corticosteroid” use during hospitalization but this is not included as a confounder. Could treatment regimen based on disease severity be driving the associations? This is particularly plausible if treatments target inflammation affecting leukocyte distribution.

Authors’ response: As suggested by the reviewer, we have performed a sensitivity analysis by adjusting for corticosteroid use and compared with the primary model. As shown in the cluster plot (please see below), the effect estimates for all 46 DMCs are highly consistent between the primary model (i.e., the one **without** adjusting for corticosteroid use) and the sensitivity model (i.e., the other with **adjusting for corticosteroid use**). This indicates that the corticosteroid use did not affect the primary association results.

4. While the single CpG analyses are interesting, regional DNA methylation analyses could enhance the biological interpretation of findings. This should be considered in the manuscript to find entire genomic regions that might relate to PPV.

Authors’ response: We thank the reviewer for the important comment. As suggested, we have conducted regional DNA methylation analysis using a commonly used *comb-p* method (Pedersen et al. 2012). The analysis has identified a total of 26 differentially methylated regions (*new*

Supplementary Table 4 below). For example, we have found that the DMR chr1: 153599487-153599831 (Šidák p-value = 1.45×10^{-3} , 11 CpGs, gene symbol: *SI00A13*) was associated with PPV use. A recent MR study found that *SI00A13* was potentially a causal regulator of IgE level (Recto et al. 2023).

As suggested, we have revised the Methods (page 18, paragraph 1) and Results (page 8, paragraph 1) sections. The text now states:

Methods section: “To identify the DMRs associated with PPV use, we applied the comb-p method to the EWAS result. Specifically, the following parameters were used in the comb-p pipeline to identify DMRs: 1) window size of 1kb (--dist 1000); 2) minimum p-value of 0.01 to start a region (--seed 0.01); 3) Šidák p-value less than 0.05; and 4) at least 3 CpGs in the region. The annotations of the DMRs, including the nearest gene and transcript, were obtained from the UCSC genome browser (hg19).”

Results section: “Additionally, in the region-based analysis, a total of 26 differentially methylated regions (DMRs) were significantly associated with the risk of PPV use (Šidák p-value < 0.05; **Supplementary Table 4**).”

References:

Pedersen BS, Schwartz DA, Yang IV, Kechris KJ. Comb-p: software for combining, analyzing, grouping and correcting spatially correlated P-values. *Bioinformatics*. 2012;28(22):2986-8.

Recto KA, Huan T, Lee DH, Lee GY, Gereige J, Yao C, Hwang SJ, Joehanes R, Kelly RS, Lasky-Su J, O'Connor G, Levy D. Transcriptome-wide association study of circulating IgE levels identifies novel targets for asthma and allergic diseases. *Front Immunol*. 2023;14:1080071.

Supplementary Table 4. Twenty-six severity-related differential methylated regions in infant hospitalized with bronchiolitis

Region_chr	start	end	no.CpGs	z_sidak_p	Transcript	GeneSymbol	distance2TSS	Promoter
chr6	3849190	3849818	22	1.48×10^{-16}	uc003mvu.3	FAM50B	0	TRUE
chr5	14326044	14326531	4	4.22×10^{-11}	uc003jfh.1	TRIO	32923	FALSE
chr5	43037123	43037666	7	3.02×10^{-10}	uc003jnf.3	ANXA2R	2781	FALSE
chr1	15272082	15272567	8	9.63×10^{-6}	uc001avq.2	KAZN	0	TRUE
chr2	220108094	220108496	7	2.07×10^{-5}	uc010zqx.2	GLB1L	0	TRUE
chr8	77913262	77913341	5	6.01×10^{-5}	uc022awe.1	PEX2	0	TRUE
chr2	27301369	27301651	6	2.20×10^{-4}	uc010eyq.2	EMILIN1	0	TRUE
chr3	122296369	122296613	8	3.97×10^{-4}	uc003efm.2	PARP15	0	TRUE
chr17	46698820	46699155	6	4.17×10^{-4}	uc002inx.3	HOXB9	4680	FALSE
chr17	40838861	40839022	3	6.53×10^{-4}	uc002iay.3	CNTNAP1	4229	FALSE
chr19	11784647	11785062	8	6.83×10^{-4}	uc021upi.1	ZNF833P	0	TRUE
chr1	203734256	203734559	6	6.84×10^{-4}	uc001haa.3	LAX1	0	TRUE
chr2	27665079	27665306	6	1.17×10^{-3}	uc002rks.3	KRTCAP3	0	TRUE
chr1	153599487	153599831	11	1.45×10^{-3}	uc001fcf.4	S100A13	0	TRUE
chr7	154684327	154684562	4	2.24×10^{-3}	uc003wlm.3	DPP6	681980	FALSE
chr13	110802517	110802968	8	2.78×10^{-3}	uc001vqw.4	COL4A1	156528	FALSE
chr10	45360781	45360969	3	2.82×10^{-3}	uc001jbk.1	TMEM72-AS1	94168	FALSE
chr22	38714166	38714466	8	3.86×10^{-3}	uc003avm.2	CSNK1E	80061	FALSE
chr10	105428385	105428651	4	4.58×10^{-3}	uc010qqs.1	SH3PXD2A	0	TRUE
chr3	3102906	3103202	4	4.73×10^{-3}	uc003bpg.3	CNTN4	21598	FALSE
chr10	93058376	93058636	3	5.59×10^{-3}	uc010qnl.2	HECTD2-AS1	312581	FALSE
chr20	1317600	1317746	3	5.74×10^{-3}	uc002wew.3	SDCBP2-AS1	11613	FALSE
chr2	128458240	128458401	7	6.33×10^{-3}	uc002tpf.3	SFT2D3	-196	TRUE
chr11	36422377	36422615	5	9.30×10^{-3}	uc010rfc.2	PRR5L	0	TRUE
chr20	31098005	31098182	3	1.12×10^{-3}	uc002wxw.1	NOLAL	26018	FALSE
chr3	112013130	112013231	3	1.20×10^{-2}	uc003dyu.3	SLC9C1	-56	TRUE

Abbreviations: chr, chromosome; CpG, cytosine-phosphate-guanine; TSS, transcription-start site

REVIEWER COMMENTS

Reviewer #1 (Remarks to the Author):

The authors were responsive to reviewer comments. They provide convincing data to support that the surrogate variables included in their primary model appropriately adjust for cell proportions. They also included another gene expression dataset. They now acknowledge additional limitations of the study pointed out by reviewers, as data are not available at this time to address them analytically.

Reviewer #2 (Remarks to the Author):

The authors have responded to all my comments and have improved the manuscript with additional analyses. My only remaining comment is that I feel the data presented showing the consistency of the effect estimates for all 46 DMs between the primary model and the sensitivity model would be of value as a supplementary figure and commenting on specifically in the discussion as it shows that although neutrophil proportions are strongly associated with PPV use, it is not neutrophil numbers driving the dmCpG associations. Of course, as the authors acknowledge their data still can't speak to causality as the association with severity might reflect, for example, activation state of neutrophils as a consequence of infection rather than any inherent epigenetic susceptibility to more severe responses to infection.

Reviewer #3 (Remarks to the Author):

Thank you for addressing the majority of concerns.

Also requested by reviewer #1 and shown by the authors on response to point 2 for reviewer #1; estimated cell-type composition is a strong predictor of PPV even after FDR adjustment. Effect sizes particularly for neutrophil composition appear large and strong in effect size. This needs to be included in the manuscript as a main finding. How much of the variance is explained by neutrophils?

The fact that neutrophils had the majority of associations (28 out of 46 CpGs) might indicate immune cell expansion from this cell sub type and not a "true" epigenetic difference. The EWAS needs to be redone adjusting for cell-type composition as estimated in EpiDish and conclusion drawn from these models.

The correlation of coefficients adjusting and not adjusting for cell-type composition only shown correlated effect sizes but EWAS conclusions are based on significant p values. This remains as a major issue of the manuscript. Given the availability of reference leukocyte composition this needs to be included as covariate for all models and presented as such for main findings (both individual CpGs and for downstream region identification).

For the conclusions based on CellDMC this method assumes all other cell-types go to zero (0). For example, the effect of B cells is estimated assuming all other cells are 0%. The authors need to comment on how valid is this method and conclusions given that the actual data is mixed cell-types and no single DNA methylation measurement contains a sample with 100% of any given cell-type. This infers that projections and conclusions are being made outside of the range of the observed obtained data.

Reviewer #1 (Remarks to the Author):

The authors were responsive to reviewer comments. They provide convincing data to support that the surrogate variables included in their primary model appropriately adjust for cell proportions. They also included another gene expression dataset. They now acknowledge additional limitations of the study pointed out by reviewers, as data are not available at this time to address them analytically.

Authors' response: We thank the reviewer for the positive comments.

Reviewer #2 (Remarks to the Author):

The authors have responded to all my comments and have improved the manuscript with additional analyses. My only remaining comment is that I feel the data presented showing the consistency of the effect estimates for all 46 DMCs between the primary model and the sensitivity model would be of value as a supplementary figure and commenting on specifically in the discussion as it shows that although neutrophil proportions are strongly associated with PPV use, it is not neutrophil numbers driving the dmCpG associations. Of course, as the authors acknowledge their data still can't speak to causality as the association with severity might reflect, for example, activation state of neutrophils as a consequence of infection rather than any inherent epigenetic susceptibility to more severe responses to infection.

Authors' response: We thank the reviewer for the positive comments. According to Reviewer #3's additional comments #2 and #3, we have re-done the EWAS analysis by additionally adjusting the seven blood cell types in the model. As a result, we have identified 33 DMCs (FDR<0.05), with 32 of them being same with the previously identified 46 DMCs. Thus, the sensitivity analysis comparing the effect estimates for all previously identified 46 DMCs between the primary model and the sensitivity model was not included in this revision.

Reviewer #3 (Remarks to the Author):

Thank you for addressing the majority of concerns.

Authors' response: We thank the reviewer for the positive comments.

1. Also requested by reviewer #1 and shown by the authors on response to point 2 for reviewer #1; estimated cell-type composition is a strong predictor of PPV even after FDR adjustment. Effect sizes particularly for neutrophil composition appear large and strong in effect size. This

needs to be included in the manuscript as a main finding. How much of the variance is explained by neutrophils?

Authors' response: We thank the reviewer for the important comment. As suggested, we have included the results of the association between the seven blood cell proportions and the risk of PPV use (*new Supplementary Table 5*) in the manuscript.

We have revised the Methods (page 19, paragraph 1) and Results (page 8, paragraph 2) sections. The text now states:

Methods section: "After estimating cell type fractions, we investigated the association of seven cell types with the risk of PPV use and ..."

Results section: "Seven blood cells types were deconvoluted and inferred. Four cell types (helper T cells, monocytes, NK cells, and neutrophils) were significantly associated with the risk of PPV use (FDR<0.05; **Supplementary Table 5**). Among them, neutrophils were the most strongly associated with the risk of PPV use (effect estimate=0.13, FDR=7.80×10⁻⁵)."

Supplementary Table 5. Association of seven blood cell types and bronchiolitis severity

Cell type	Effect Estimate	SE	P	FDR
B cells	-0.02	0.01	0.12	0.14
NK cells	-0.01	0.01	0.01	0.01
Helper T cells	-0.06	0.02	0.00	0.01
Cytotoxic T cells	-0.02	0.01	0.06	0.08
Monocytes	-0.02	0.01	0.01	0.01
Neutrophils	0.13	0.03	0.00	7.80×10 ⁻⁵
Eosinophils	-0.01	0.00	0.14	0.14

Lastly, we have calculated the variance explained by each cell type. As expected, neutrophils had the highest variance (0.0264) comparing with other cell types.

Cell types	Variance
B cells	0.0035
NK cells	0.0008
Helper T cells	0.0104
Cytotoxic T cells	0.0021
Monocytes	0.0016
Neutrophils	0.0264
Eosinophils	0.0004

2. The fact that neutrophils had the majority of associations (28 out of 46 CpGs) might indicate immune cell expansion from this cell sub type and not a "true" epigenetic difference. The EWAS needs to be redone adjusting for cell-type composition as estimated in EpiDish and conclusion drawn from these models.

Authors' response: We thank the reviewer for the important comments. As suggested, we have re-done the EWAS analysis by additionally adjusting the seven blood cell types in the model. As a result, we have identified 33 DMCs (FDR<0.05), with 32 of them being same with the previously identified 46 DMCs. In addition, in the new region-based analysis, a total of 22 DMRs were significantly associated with the risk of PPV use (Šidák p-value <0.05), with 15 of them being same with the previously identified 26 DMRs. Based on the new EWAS results, we have also re-done most of the downstream analysis when it is applicable. Thus, **Table 2, Figures 1-4, Supplementary Tables 4 and 6, Supplementary Figure 4**, were updated.

We have revised the Methods and Results sections accordingly. Most importantly, we have revised the covariate adjustment in the Methods section (page 19, paragraph 1). The text now states:

“In the EWAS analysis, we adjusted for potential confounders, including age, sex, race/ethnicity, number of previous breathing problems, RSV infection, prematurity, seven blood cell types (B cells, T_C cells, T_H cells, eosinophils, monocytes, neutrophils, and NK cells) and the derived surrogate variables based on a priori knowledge and clinical plausibility.”

Lastly, we have also uploaded the new bronchiolitis severity EWAS summary statistics at our research website <http://lianglab.rc.fas.harvard.edu/BronchiolitisSeverityEWAS/>.

3. The correlation of coefficients adjusting and not adjusting for cell-type composition only shown correlated effect sizes but EWAS conclusions are based on significant p values. This remains as a major issue of the manuscript. Given the availability of reference leukocyte composition this needs to be included as covariate for all models and presented as such for main findings (both individual CpGs and for downstream region identification).

Authors' response: We thank the reviewer for the comments. As suggested, we have re-done the EWAS analysis by additionally adjusting the seven blood cell types in the model. For details, please see our response to the previous comment #2.

4. For the conclusions based on CellDMC this method assumes all other cell-types go to zero (0). For example, the effect of Bcells is estimated assuming all other cells are 0%. The authors need

to comment on how valid is this method and conclusions given that the actual data is mixed cell-types and no single DNA methylation measurement contains a sample with 100% of any given cell-type. This infers that projections and conclusions are being made outside of the range of the observed obtained data.

Authors' response: We thank and agree with the reviewer for the important comments. As suggested, we have highlighted this potential limitation in the Discussion section (page 15, paragraph 1). The text states:

“Fifth, the results of DMCs in each cell type need to be interpreted with caution since “CellDMC” function in the *EpiDISH* package assumes all other cell types are 0% when it estimates a specific cell type driving the methylation change, where our data contain mixed cell types”.

REVIEWERS' COMMENTS

Reviewer #3 (Remarks to the Author):

The authors have sufficiently address the limitations and analyses adjusted for cell-types are now interpreted.

Reviewer #3 (Remarks to the Author):

The authors have sufficiently address the limitations and analyses adjusted for cell-types are now interpreted.

Authors' response: We thank the reviewer for the positive comment.